# From Deterministic to Probabilistic World: Balancing Enhanced Doubly Robust Learning for Debiased Recommendations

## Abstract

In recommender systems, selection bias arises from the users' selective interactions with items, which poses a widely-recognized challenge for unbiased evaluation and learning for recommendation models. Recently, doubly robust and its variants have been widely studied to achieve debiased learning of prediction models, which enables unbiasedness when either imputed errors or learned propensities are accurate. However, we find that previous studies achieve unbiasedness using the doubly robust learning approaches are all based on deterministic error imputation model and deterministic propensity model, and these approaches fail to be unbiased when using probabilistic models to impute errors and learn propensities. To tackle this problem, in this paper, we first derive the bias of doubly robust learning methods and provide alternative unbiasedness conditions for probabilistic models. Then we propose a novel balancing enhanced doubly robust joint learning approach, which improves the accuracy of the imputed errors and leads to unbiased learning under probabilistic error imputations and learned propensities. We further derive the generalization error bound when using the probabilistic models, and show that it can be effectively controlled by the proposed learning approach. We conduct extensive experiments on three real-world datasets, including a large-scale industrial dataset, to demonstrate the effectiveness of the proposed method.

## 1 Introduction

In the era of information explosion, recommender system (RS) plays an increasingly important role in e-commerce platforms, social media, news reading, and other places. However, due to the subjective preferences of users and the data collection process itself, selection bias always exists in the collected data (Pradel et al., 2012), which poses a widely-recognized challenge (De Myttenaere et al., 2014; Marlin and Zemel, 2009). Ignoring selection bias makes RS difficult to provide quality and accurate recommended content to users, thus hurting the users experience and reducing social benefits.

Many methods have been proposed to address selection bias. The error imputation based (EIB) method (Chang et al., 2010; Marlin et al., 2007; Steck, 2010; 2013) utilizes imputation model to impute the missing relevance. However, it is very difficult to obtain an accurate imputed value for missing events due to the data sparsity and the covariate shift in practice (Dudík et al., 2011; Marlin et al., 2007; Steck, 2013). The inverse propensity score (IPS) method uses inverse propensity to reweight the observed events to achieve unbiasedness (Imbens and Rubin, 2015; Saito et al., 2020; Schnabel et al., 2016). Nevertheless, the accurate propensities are difficult to be accurately estimated and includes extremely small values, which brings a large variance to the IPS method to lead the poor generalizatin performance (Rosenbaum and Rubin, 1983; Swaminathan and Joachims, 2015; Thomas and Brunskill, 2016). The doubly robust (DR) method combines the error imputation model and the propensity model (Wang et al., 2019; Saito, 2020; Wang et al., 2022; Oosterhuis, 2023), which is unbiased if either the imputed errors or the learned propensities are accurate, which is also proved to has smaller variance compared to the IPS method (Saito, 2020; Oosterhuis, 2022; Li et al., 2023c).

Despite the broad use of the error imputation model and the propensity model in the above methods, we find that these methods are limited to deterministic error imputation model and deterministic propensity model, which fails to be unbiased when using probabilistic models to impute errors and

learn propensities. One of the most significant advantages of the probabilistic modeling technique is that it provides a comprehensive understanding of the uncertainty associated with the imputed errors and the learned propensities (Murphy, 2012). In this way, we can effectively determine how confident the imputation and propensity models are and how accurate its prediction is (Ghahramani, 2015; Murphy, 2022). Therefore, it is essential to extend the previously widely adopted DR methods to be compatible with probabilistic error imputation and propensity models.

To this end, in this paper, we first derive the bias of doubly robust learning methods and provide alternative unbiasedness conditions for probabilistic error imputations and propensity estimations. Then we propose a novel balancing enhanced doubly robust (BEDR) joint learning approach, which improves the accuracy of the imputed errors and leads to unbiased learning under probabilistic error imputations and learned propensities. We further derive the generalization error bound when using the probabilistic models, and show that it can be effectively controlled by the proposed learning approach. Extensive experiments are conducted on three real-world datasets, including a large-scale industrial dataset, to demonstrate the effectiveness of the proposed method.

Our main contributions can be summarized as follows:

- To the best of our knowledge, this is the first paper that extends previous widely adopted DR methods to be compatible with probabilistic error imputation and propensity models. We show the bias of the DR estimator under probabilistic models has two terms: a covariance term and a term measured the accuracy of learned propensities and imputed errors.

- In order to control the covariance term while obtaining accurate learned propensities and imputed errors, we make subtle changes to the previous DR method and propose a simple and effective BEDR method, which is unbiased under probabilistic models.

- Our theoretical analysis further shows that controlling the covariance term not only achieves unbiased estimation but also reduces the generalization bound to enhance the prediction performance under probabilistic learned propensities and imputed errors.

- We conduct extensive experiments on three real-world datasets, including a large industrial dataset, to demonstrate the effectiveness of our proposed method.

## 2 RELATED WORK

There are various biases in the data collected from RS (Chen et al., 2020; Wu et al., 2022), which have been of increasing concern in recent years (Ai et al., 2018; Saito and Nomura, 2022; Liu et al., 2021; Zhang et al., 2021; Luo et al., 2021; Liu et al., 2022; Lin et al., 2023). Selection bias is one of the most common bias in RS and a lot of research has been done aiming to eliminate this kind of bias (Chen et al., 2021; Guo et al., 2021; Liu et al., 2020; Saito, 2020; Schnabel et al., 2016; Wang et al., 2019). The error imputation based method (EIB) (Chang et al., 2010; Marlin et al., 2007; Steck, 2010; 2013) first imputes pseudo-labels for missing events from the observed events, and then combines the pseudo-labels with the observed events to train the prediction model (Dudík et al., 2011; Marlin et al., 2007; Steck, 2013; Wu et al., 2022). An alternative way to eliminate selection bias is to weight the inverse propensity score (IPS) on the observed data to eliminate bias (Imbens and Rubin, 2015; Saito et al., 2020; Schnabel et al., 2016). However, IPS will suffer from a large variance when the extreme values exist in the estimated propensities (Li et al., 2023c; Thomas and Brunskill, 2016). Doubly robust (DR) method improves the weakness of EIB and IPS methods and becomes the mainstream model due to the weaker unbiasedness conditions and smaller variance than the IPS method (Benkeser et al., 2017; Morgan and Winship, 2015; Luo et al., 2021; Li et al., 2023c; Saito, 2020; Wang et al., 2019). In particular, DR estimator is unbiased when either learned propensities or imputed errors are accurate. Many augmented DR methods are developed to further enhance the previous DR method performance by modifying the propensity model and imputation model or the form of the DR estimator, such as MRDR (Guo et al., 2021), BRD-DR (Ding et al., 2022), StableDR (Li et al., 2023c), TDR (Li et al., 2023b), DR-MSE (Dai et al., 2022), and DR-BIAS (Dai et al., 2022). However, these approaches are limited to the using of deterministic error imputation and propensity models, and fail to be unbiased when using probabilistic models to impute errors and learn propensities. To the best of our knowledge, this is the first paper that extends previous widely adopted DR methods to be compatible with probabilistic error imputation and propensity models.

## 3 Preliminaries

Suppose the user set $\mathcal{U} = \{u_1, u_2, \ldots, u_m\}$ contains $m$ users, the item set $\mathcal{I} = \{i_1, i_2, \ldots, i_n\}$ contains $n$ items, and denote the set of all user-item pairs as $\mathcal{D} = \mathcal{U} \times \mathcal{I}$. Let $\mathbf{R} \in \mathbb{R}^{m \times n}$ be the ground truth rating matrix of all user-item pairs, where $r_{u,i}$ is the rating of user $u$ on item $i$. Let $x_{u,i}$ be the feature of user $u$ and item $i$, and $\hat{r}_{u,i} = f(x_{u,i}; \theta)$ is the predicted rating by a prediction model, $\theta$ is the corresponding parameter. Denote $\hat{\mathbf{R}} \in \mathbb{R}^{m \times n}$ be the matrix contains all the predicted ratings. Let $\mathbf{O}^{m \times n}$ be the binary observation indicator matrix for all user-item pair, $o_{u,i} = 1$ indicates the rating of user $u$ on item $i$ is observed, otherwise missing $o_{u,i} = 0$. The purpose of RS is to train a prediction model to accurately predict all ratings. If all the ratings are observed, the prediction model can be trained directly by minimizing the following ideal loss function:

$$\mathcal{L}_{ideal}(\theta) = \frac{1}{|\mathcal{D}|} \sum_{(u,i) \in \mathcal{D}} e_{u,i},$$

where $e_{u,i} = \mathcal{L}(\hat{r}_{u,i}, r_{u,i})$ is the loss between the predicted rating $\hat{r}_{u,i}$ and the ground truth rating $r_{u,i}$ and $\mathcal{L}(\cdot, \cdot)$ is an arbitrary loss function. However, the ideal loss function is not available in most cases because we can only observe partial biased data. To tackle this issue, many unbiased estimators of the ideal loss have been proposed. For example, the EIB estimator imputes the missing $e_{u,i}$ with the imputed errors $\hat{e}_{u,i}$ as shown below:

$$\mathcal{E}_{EIB}(\theta) = \frac{1}{|\mathcal{D}|} \sum_{(u,i) \in \mathcal{D}} o_{u,i} e_{u,i} + (1 - o_{u,i}) \hat{e}_{u,i},$$

where $\hat{e}_{u,i} = m(x_{u,i}; \phi)$ is the error imputation model for estimating $e_{u,i}$ using $x_{u,i}$. In addition, the IPS estimator uses inverse propensity to reweight the observed events, as shown below:

$$\mathcal{E}_{IPS}(\theta) = \frac{1}{|\mathcal{D}|} \sum_{(u,i) \in \mathcal{D}} \frac{o_{u,i} e_{u,i}}{\hat{p}_{u,i}},$$

where $\hat{p}_{u,i} = \pi(x_{u,i}; \psi)$ is the propensity model for estimating $p_{u,i} := \mathbb{P}(o_{u,i} = 1 \mid x_{u,i})$. Moreover, the DR estimator combines the $\hat{p}_{u,i}$ and $\hat{e}_{u,i}$ together, which is constructed as:

$$\mathcal{E}_{DR}(\theta) = \frac{1}{|\mathcal{D}|} \sum_{(u,i) \in \mathcal{D}} \left[ \hat{e}_{u,i} + \frac{o_{u,i}(e_{u,i} - \hat{e}_{u,i})}{\hat{p}_{u,i}} \right],$$

In below, we focus on the theoretical properties of the DR estimator, and start from the widely-known conclusions of DR estimator under deterministic imputed errors and propensities.

**Lemma 1** (Bias of DR Estimator under **Deterministic Models** (Wang et al., 2019)). *Given imputed errors $\hat{e}_{u,i}$ and learned propensities $\hat{p}_{u,i} > 0$ for all user-item pairs, when considering only the randomness of rating missing indicators, the bias of the DR estimator is*

$$\text{Bias}_{\mathcal{O}}[\mathcal{E}_{DR}(\theta)] = \frac{1}{|\mathcal{D}|} \sum_{(u,i) \in \mathcal{D}} \frac{\{\hat{p}_{u,i} - p_{u,i}\} \cdot \{e_{u,i} - \hat{e}_{u,i}\}}{\hat{p}_{u,i}}.$$

We find that either $\hat{e}_{u,i} = e_{u,i}$ or $\hat{p}_{u,i} = p_{u,i}$ is sufficient to eliminate bias under deterministic models, which inspires the double robustness condition for the DR method.

**Corollary 1** (Double Robustness under **Deterministic Models** (Wang et al., 2019)). *The DR estimator is unbiased when either imputed errors $\hat{e}_{u,i}$ or learned propensities $\hat{p}_{u,i}$ are accurate for all user-item pairs, i.e., either $\hat{e}_{u,i} = e_{u,i}$ or $\hat{p}_{u,i} = p_{u,i}$.*

## 4 Proposed Method

### 4.1 Motivation

The above lemma 1 shows the bias form of DR estimator under deterministic error imputation and propensity models. However, such theoretical results no longer hold when using the probabilistic models to impute the prediction errors and learn propensities. The following theorem shows the bias form of the DR estimator under probabilistic error imputation and propensity models.

**Theorem 1** (Bias of DR Estimator under **Probabilistic Models**). *Given probabilistic error imputation model $\mathbb{E}(\hat{e}_{u,i} \mid x_{u,i})$ and probabilistic propensity model $\mathbb{P}(\hat{p}_{u,i} \mid x_{u,i})$, consider all variables are random, then the bias of the DR estimator, namely $\mathrm{Bias}[\mathcal{E}_{DR}(\theta)]$, is*

$$\underbrace{\mathrm{Cov}\left(\frac{\hat{p}_{u,i} - o_{u,i}}{\hat{p}_{u,i}}, e_{u,i} - \hat{e}_{u,i}\right)}_{\textit{equals to 0 if independent}} + \underbrace{\mathbb{E}\left[\left\{1 - \mathbb{E}\left[\frac{o_{u,i}}{\hat{p}_{u,i}} \Big| x_{u,i}\right]\right\} \cdot \{\mathbb{E}[e_{u,i} \mid x_{u,i}] - \mathbb{E}[\hat{e}_{u,i} \mid x_{u,i}]\}\right]}_{\textit{equals to 0 either } \mathbb{E}[o_{u,i}/\hat{p}_{u,i} \mid x_{u,i}] = 1 \textit{ or } \mathbb{E}[\hat{e}_{u,i} - e_{u,i} \mid x_{u,i}] = 0}$$

**Corollary 2** (Double Robustness under **Probabilistic Models**). *The DR estimator is unbiased when both the following conditions hold:*

*(i) the covariance term vanishes, i.e., $\mathrm{Cov}((\hat{p}_{u,i} - o_{u,i})/\hat{p}_{u,i}, e_{u,i} - \hat{e}_{u,i}) = 0$;*

*(ii) either learned propensities satisfy $\mathbb{E}[o_{u,i}/\hat{p}_{u,i} \mid x_{u,i}] = 1$, or imputed errors have the same conditional expectation with true prediction errors $\mathbb{E}[\hat{e}_{u,i} \mid x_{u,i}] = \mathbb{E}[e_{u,i} \mid x_{u,i}]$.*

Compared with the existing theoretical results as in lemma 1, it is intuitive that condition *(ii)* is necessary to achieve unbiasedness, which directly extends the conditions of accurate imputed errors and learned propensities in lemma 1 to the probabilistic form. However, it should be noted that the condition *(i)* that covariance vanish is also needed for the unbiasedness under probabilistic models. Therefore, it is necessary to modify the previous DR learning approaches to be compatible with probabilistic error imputation and propensity models.

## 4.2 THE BALANCING ENHANCED DR ESTIMATOR

It is important to note that the true covariance is unknown due to the fact that the true distribution of the data is unknown. However, we can use the empirical covariance over all user-item pairs as an approximate of the true covariance. We first give the definition of empirical covariance.

**Definition 1** (Empirical Covariance). *The empirical expected conditional covariance between $(\hat{p}_{u,i} - o_{u,i})/\hat{p}_{u,i}$ and $e_{u,i} - \hat{e}_{u,i}$ is*

$$\widehat{\mathrm{Cov}}\left(\frac{\hat{p}_{u,i} - o_{u,i}}{\hat{p}_{u,i}}, e_{u,i} - \hat{e}_{u,i}\right) = \frac{1}{|\mathcal{D}|} \sum_{(u,i) \in \mathcal{D}} \frac{\hat{p}_{u,i} - o_{u,i}}{\hat{p}_{u,i}} \cdot (e_{u,i} - \hat{e}_{u,i}).$$

A direct method to control the empirical covariance is to regard it as a regularization term. However, since the data are partially observed, we cannot obtain the value of the empirical covariance on all user-item pairs. In addition, doing so cannot guarantee the empirical covariance converges to exact zero, and the large penalty term may hurt the prediction performance. Interestingly, we found that the empirical covariance can be controlled with subtle changes to the DR estimator. Specifically, we designed imputation balancing correction as follows:

$$\tilde{e}_{u,i} = m(x_{u,i}; \phi) + \epsilon(o_{u,i} - \pi(x_{u,i}; \psi)).$$

Motivated by targeted maximum likelihood estimation (van der Laan and Rose, 2011), we add a correction term $\epsilon(o_{u,i} - \pi(x_{u,i}; \psi))$ on $\hat{e}_{u,i}$, which has zero mean under accurate $\pi(x_{u,i}; \psi)$, thus will not bring extra bias to the imputation model. We then learn $\phi$ and $\epsilon$ in $\tilde{e}_{u,i}$ by minimizing

$$(\phi^*, \epsilon^*) = \arg\min_{\phi,\epsilon} \mathcal{L}_e^{Bal}(\phi, \epsilon) = \frac{1}{|\mathcal{D}|} \sum_{(u,i) \in \mathcal{D}} \frac{o_{u,i}(e_{u,i} - \tilde{e}_{u,i})^2}{\hat{p}_{u,i}} + v\|\phi\|_F^2,$$

where $\|\cdot\|_F^2$ is the Frobenius norm. This proposed loss has several desired properties. First, the derivatives on the proposed loss with respect to $\epsilon$ is shown below:

$$\frac{\partial}{\partial \epsilon} \mathcal{L}_e^{Bal}(\phi, \epsilon) = \frac{2}{|\mathcal{D}|} \sum_{(u,i) \in \mathcal{O}} \frac{\hat{p}_{u,i} - o_{u,i}}{\hat{p}_{u,i}} \cdot (e_{u,i} - \hat{e}_{u,i}).$$

It has the same form as the empirical covariance for user-item pairs with $o_{u,i} = 1$, which means that we can make the empirical covariance for observed user-item pairs to exact zero by minimizing the $\mathcal{L}_e^{Bal}$ directly. Meanwhile, the gradient contains $\epsilon$ when taking the derivatives with respect to $\phi$, which indicates a well-learned $\epsilon$ can lead to a more accurate $\phi$ to further ensure unbiasedness. Moreover, the unobserved empirical covariance can also be bounded by $\mathcal{L}_e^{Bal}$ using the concentration inequality. The theorem 2 below formally shows the controllability of empirical covariance.

**Theorem 2** (Controllability of Empirical Covariance). *The boosted imputation model trained by minimizing the balanced enhanced imputation loss is sufficient for controlling the empirical covariance.*

*(i) For user-item pairs with **observed** outcomes, the empirical covariance is 0. Formally, we have*

$$\frac{\partial}{\partial \epsilon} \mathcal{L}_e^{Bal}(\phi, \epsilon)\Big|_{\epsilon = \epsilon^*} = 0, \quad \text{which is equivalent to} \quad \frac{1}{|\mathcal{D}|} \sum_{(u,i): \, o_{u,i}=1} \frac{\hat{p}_{u,i} - o_{u,i}}{\hat{p}_{u,i}} \cdot (e_{u,i} - \tilde{e}_{u,i}) = 0;$$

*(ii) For user-item pairs with **missing** outcomes, suppose that $\hat{p}_{u,i} \geq K_\psi$ and $|e_{u,i} - \tilde{e}_{u,i}| \leq K_\phi$, then with probability at least $1 - \eta$, we have*

$$\frac{1}{|\mathcal{D}|} \sum_{(u,i): \, o_{u,i}=0} \frac{\hat{p}_{u,i} - o_{u,i}}{\hat{p}_{u,i}} \cdot (e_{u,i} - \tilde{e}_{u,i}) \leq \underbrace{\mathcal{L}_e^{Bal}(\phi, \epsilon)^{\frac{1}{2}}}_{\text{proposed loss}} + K_\phi \cdot \underbrace{\left[ \frac{1}{|\mathcal{D}|} \sum_{u,i \in \mathcal{D}} \left| 1 - \mathbb{E}\left[ \frac{o_{u,i}}{\hat{p}_{u,i}} \Big| x_{u,i} \right] \right| \right]^{\frac{1}{2}}}_{\text{empirical bias from probabilistic propensity model}}$$

$$+ \underbrace{\left[ K_\phi \left( 1 + \frac{1}{K_\psi} \right) \left( 2\mathcal{R}(\mathcal{F}) + (2K_\phi + 1)\sqrt{\frac{2\log(4/\eta)}{|\mathcal{D}|}} \right) \right]^{\frac{1}{2}}}_{\text{tail bound controlled by empirical Rademacher complexity and sample size}}.$$

It should be noted that the proposed imputation balancing correction has no harm property. That is, when the previous $\hat{e}_{u,i}$ has already ensure the empirical covariance to zero, the $\epsilon$ will converge to zero to degrade, as shown in the following remark 1.

*Remark* 1 (Relation to previous imputed errors). The learned coefficient $\epsilon^*$ will converge to zero, when the probabilistic imputation model $\hat{e}_{u,i}$ has already lead to zero empirical covariance, making $\tilde{e}_{u,i}$ degenerates to $\hat{e}_{u,i}$.

In addition, the remark 2 below shows that the proposed imputation balancing correction can not only control the empirical covariance efficiently, but also be helpful for learning more accurate imputed errors when the previous imputed errors are inaccurate.

*Remark* 2 (Bias reduction property). The proposed balancing enhanced imputation loss leads to smaller bias of imputed errors $\tilde{e}_{u,i}$, when $\hat{e}_{u,i}$ are inaccurate. Formally,

$$\min_{\phi, \epsilon} \mathcal{L}_e^{Bal}(\phi, \epsilon) = \frac{1}{|\mathcal{D}|} \sum_{(u,i) \in \mathcal{D}} \frac{o_{u,i}(e_{u,i} - \tilde{e}_{u,i})^2}{\hat{p}_{u,i}} \leq \min_\phi \mathcal{L}_e(\phi) = \frac{1}{|\mathcal{D}|} \sum_{(u,i) \in \mathcal{D}} \frac{o_{u,i}(e_{u,i} - \hat{e}_{u,i})^2}{\hat{p}_{u,i}}.$$

Moreover, while reducing bias, the proposed method also reduces the variance compared to the previous imputed errors under a moderate condition, as shown in the remark 3 below.

*Remark* 3 (Variance reduction property). The proposed balancing enhanced imputation loss leads to smaller variance of imputed errors $\tilde{e}_{u,i}$ when the optimal $\epsilon^*$ lies in a certain range. Formally, we have

$$\mathbb{V}(\tilde{e}_{u,i}) = \mathbb{V}(\hat{e}_{u,i} + \epsilon^* \cdot (o_{u,i} - \hat{p}_{u,i})) \leq \mathbb{V}(\hat{e}_{u,i}), \quad \text{if} \quad \epsilon^* \in \left[ 0, \, 2 \cdot \frac{\text{Cov}(\hat{e}_{u,i}, \hat{p}_{u,i} - o_{u,i})}{\mathbb{V}(\hat{p}_{u,i} - o_{u,i})} \right].$$

Finally, the proposed BEDR estimator is given as:

$$\mathcal{E}_{BEDR}(\theta) = \frac{1}{|\mathcal{D}|} \sum_{(u,i) \in \mathcal{D}} \left[ \tilde{e}_{u,i} + \frac{o_{u,i}(e_{u,i} - \tilde{e}_{u,i})}{\hat{p}_{u,i}} \right],$$

where $\tilde{e}_{u,i} = m(x_{u,i}; \phi) + \epsilon(o_{u,i} - \pi(x_{u,i}; \psi))$.

### 4.3 THE LEARNING ALGORITHM

We optimize the prediction model and the imputation model of the BEDR method by a widely used joint learning framework (Wang et al., 2019), which alternatively optimize two model to achieve unbiased learning. Specifically, we train the prediction model by minimizing the BEDR training loss:

$$\mathcal{L}_{BEDR}(\theta) = \frac{1}{|\mathcal{D}|} \sum_{(u,i) \in \mathcal{D}} \left[ \tilde{e}_{u,i} + \frac{o_{u,i}(e_{u,i} - \tilde{e}_{u,i})}{\hat{p}_{u,i}} \right] + v\|\theta\|_F^2,$$

Then we update the imputation model parameters and $\epsilon$ simultaneously by minimizing the $\mathcal{L}_e^{Bal}(\phi, \epsilon)$ in Section 4.2. The parameters of the prediction and imputation model are updated alternatively via minbatch stochastic gradient descent. We summarize the joint learning process in Algorithm 1.

---

**Algorithm 1:** The Proposed **Balancing Enhanced** Doubly Robust Joint Learning

---

**Input:** observed ratings $\mathbf{R}^o$ and a pre-trained probabilistic propensity model $\pi(x_{u,i}; \psi)$.

1 **while** *stopping criteria is not satisfied* **do**
2      **for** *number of steps for training the balancing enhanced imputation model* **do**
3          Sample a batch of user-item pairs $\{(u_j, i_j)\}_{j=1}^J$ from $\mathcal{O}$;
4          Update $\phi$ by descending along the gradient $\nabla_\phi \mathcal{L}_e^{Bal}(\phi, \epsilon)$;
5          **Update $\epsilon$ by descending along the gradient $\nabla_\epsilon \mathcal{L}_e^{Bal}(\phi, \epsilon)$;**
6      **end**
7      **for** *number of steps for training the debiased prediction model* **do**
8          Sample a batch of user-item pairs $\{(u_k, i_k)\}_{k=1}^K$ from $\mathcal{D}$;
9          Update $\theta$ by descending along the gradient $\nabla_\theta \mathcal{L}_{BEDR}(\theta; \phi, \psi)$;
10      **end**
11 **end**

---

### 4.4 THE GENERALIZATION BOUND

Next, we analyze the generalization error bound of the DR methods using the probabilistic models for estimating $e_{u,i}$ and $p_{u,i}$, and show that controlling empirical covariance leads to a tighter bound.

**Definition 2** (Empirical Rademacher Complexity (Shalev-Shwartz and Ben-David, 2014)). *Let $\mathcal{F}$ be a family of prediction models mapping from $x \in \mathcal{X}$ to $[a, b]$, and $S = \{x_{u,i} \mid (u, i) \in \mathcal{D}\}$ a fixed sample of size $|\mathcal{D}|$ with elements in $\mathcal{X}$. Then, the empirical Rademacher complexity of $\mathcal{F}$ with respect to the sample $S$ is defined as:*

$$\mathcal{R}(\mathcal{F}) = \mathbb{E}_{\boldsymbol{\sigma} \sim \{-1, +1\}^{|\mathcal{D}|}} \sup_{f_\theta \in \mathcal{F}} \left[ \frac{1}{|\mathcal{D}|} \sum_{(u,i) \in \mathcal{D}} \sigma_{u,i} e_{u,i} \right],$$

*where $\boldsymbol{\sigma} = \{\sigma_{u,i} : (u, i) \in \mathcal{D}\}$, and $\sigma_{u,i}$ are independent uniform random variables taking values in $\{-1, +1\}$. The random variables $\sigma_{u,i}$ are called Rademacher variables.*

**Theorem 3** (Generalization Bound under **Probabilistic Models**). *Suppose that $\hat{p}_{u,i} \geq K_\psi$ and $|e_{u,i} - \hat{e}_{u,i}| \leq K_\phi$, then with probability at least $1 - \eta$, we have*

$$\mathcal{L}_{ideal}(\theta) \leq \underbrace{\mathcal{L}_{DR}(\theta) + \frac{1}{|\mathcal{D}|} \sum_{(u,i) \in \mathcal{D}} \left| 1 - \mathbb{E}\left[ \frac{o_{u,i}}{\hat{p}_{u,i}} \Big| x_{u,i} \right] \right| \cdot \left| \mathbb{E}[e_{u,i} \mid x_{u,i}] - \mathbb{E}[\hat{e}_{u,i} \mid x_{u,i}] \right|}_{\text{vanilla DR only controls the empirical DR loss, and empirical risks of imputation and propensity models}}$$

$$+ \underbrace{\left| \frac{1}{|\mathcal{D}|} \sum_{(u,i) \in \mathcal{D}} \text{Cov}\left( \frac{o_{u,i} - \hat{p}_{u,i}}{\hat{p}_{u,i}}, e_{u,i} - \hat{e}_{u,i} \right) \right|}_{\text{balancing enhanced DR further controls the independence}} + \underbrace{\left( 1 + \frac{1}{K_\psi} \right) \left( 2\mathcal{R}(\mathcal{F}) + K_\phi \sqrt{\frac{18 \log(4/\eta)}{|\mathcal{D}|}} \right)}_{\text{tail bound controlled by empirical Rademacher complexity and sample size}}$$

## 5 EXPERIMENTS

### 5.1 EXPERIMENTAL SETUP

**Dataset and Preprocessing.** To verify the effectiveness of the proposed method in the real-world dataset, the dataset that contains both biased and unbiased data is required. Following the previous studies (Saito, 2020; Wang et al., 2019; 2021; Chen et al., 2021), the following three widely used real-world datasets are adopted to conduct our experiments: **Coat** [1] contains ratings from 290 users to 300 items. Each user rates 24 of the coats that are selected by themselves, which produce 6,960 biased ratings in total. Meanwhile, each user is asked to rate 16 randomly picked items, which generates 4,640 unbiased ratings. **Yahoo! R3** [2] contains ratings from 15,400 users to 1,000 items. Each user

---

[1] https://www.cs.cornell.edu/~schnabts/mnar/
[2] http://webscope.sandbox.Music.com/

Table 1: Mean and standard deviation of the MSE, AUC, Recall@5 and NDCG@5 on **Coat** and **Yahoo! R3** datasets. The best results are bolded, and the best baseline results are underlined. $*$ means statistically significant (p-value $\leq 0.01$) using the paired-t-test compared with the best baseline.

| Method | Coat | | | | Yahoo! R3 | | | |
|---|---|---|---|---|---|---|---|---|
| | MSE $\downarrow$ | AUC $\uparrow$ | Recall@5 $\uparrow$ | NDCG@5 $\uparrow$ | MSE $\downarrow$ | AUC $\uparrow$ | Recall@5 $\uparrow$ | NDCG@5 $\uparrow$ |
| PMF | $0.239_{\pm 0.003}$ | $0.705_{\pm 0.006}$ | $0.435_{\pm 0.008}$ | $0.616_{\pm 0.009}$ | $0.250_{\pm 0.001}$ | $0.673_{\pm 0.001}$ | $0.393_{\pm 0.002}$ | $0.633_{\pm 0.002}$ |
| + IPS | $0.211_{\pm 0.001}$ | $0.715_{\pm 0.005}$ | $0.438_{\pm 0.010}$ | $0.626_{\pm 0.010}$ | $0.231_{\pm 0.002}$ | $0.684_{\pm 0.001}$ | $0.424_{\pm 0.001}$ | $0.645_{\pm 0.002}$ |
| + SNIPS | $0.226_{\pm 0.005}$ | $0.716_{\pm 0.006}$ | $0.448_{\pm 0.013}$ | $0.631_{\pm 0.011}$ | $\underline{0.201}_{\pm 0.001}$ | $0.685_{\pm 0.001}$ | $0.412_{\pm 0.002}$ | $0.637_{\pm 0.003}$ |
| + ASIPS | $0.215_{\pm 0.004}$ | $0.718_{\pm 0.010}$ | $0.443_{\pm 0.010}$ | $0.620_{\pm 0.010}$ | $0.234_{\pm 0.001}$ | $0.679_{\pm 0.001}$ | $0.422_{\pm 0.002}$ | $0.640_{\pm 0.003}$ |
| + DR | $0.238_{\pm 0.003}$ | $0.728_{\pm 0.013}$ | $0.440_{\pm 0.017}$ | $0.636_{\pm 0.019}$ | $0.208_{\pm 0.001}$ | $0.686_{\pm 0.003}$ | $0.421_{\pm 0.002}$ | $0.657_{\pm 0.002}$ |
| + DR-JL | $0.210_{\pm 0.001}$ | $0.735_{\pm 0.008}$ | $0.444_{\pm 0.005}$ | $0.630_{\pm 0.004}$ | $0.218_{\pm 0.001}$ | $0.686_{\pm 0.002}$ | $0.428_{\pm 0.003}$ | $0.650_{\pm 0.003}$ |
| + CVIB | $0.225_{\pm 0.001}$ | $0.730_{\pm 0.003}$ | $0.437_{\pm 0.006}$ | $0.633_{\pm 0.007}$ | $0.256_{\pm 0.001}$ | $0.683_{\pm 0.001}$ | $0.406_{\pm 0.001}$ | $0.643_{\pm 0.002}$ |
| + MRDR | $0.211_{\pm 0.002}$ | $0.732_{\pm 0.006}$ | $0.449_{\pm 0.004}$ | $0.636_{\pm 0.006}$ | $0.217_{\pm 0.001}$ | $0.686_{\pm 0.002}$ | $\underline{0.432}_{\pm 0.002}$ | $0.654_{\pm 0.002}$ |
| + DIB | $0.242_{\pm 0.001}$ | $0.737_{\pm 0.003}$ | $0.446_{\pm 0.008}$ | $\underline{0.643}_{\pm 0.009}$ | $0.248_{\pm 0.001}$ | $0.687_{\pm 0.001}$ | $0.425_{\pm 0.001}$ | $0.640_{\pm 0.002}$ |
| + DR-BIAS | $0.210_{\pm 0.003}$ | $0.727_{\pm 0.004}$ | $0.451_{\pm 0.010}$ | $0.633_{\pm 0.009}$ | $0.216_{\pm 0.001}$ | $0.686_{\pm 0.001}$ | $0.430_{\pm 0.001}$ | $0.653_{\pm 0.002}$ |
| + DR-MSE | $0.210_{\pm 0.003}$ | $\underline{0.739}_{\pm 0.004}$ | $0.446_{\pm 0.009}$ | $0.634_{\pm 0.009}$ | $0.217_{\pm 0.001}$ | $0.684_{\pm 0.003}$ | $0.429_{\pm 0.001}$ | $0.653_{\pm 0.003}$ |
| + MR | $0.211_{\pm 0.002}$ | $0.731_{\pm 0.004}$ | $0.452_{\pm 0.005}$ | $0.644_{\pm 0.010}$ | $0.247_{\pm 0.001}$ | $\underline{0.690}_{\pm 0.003}$ | $0.409_{\pm 0.003}$ | $0.647_{\pm 0.003}$ |
| + TDR | $0.221_{\pm 0.003}$ | $0.719_{\pm 0.006}$ | $0.446_{\pm 0.009}$ | $0.630_{\pm 0.010}$ | $0.225_{\pm 0.001}$ | $0.687_{\pm 0.002}$ | $0.430_{\pm 0.004}$ | $\underline{0.661}_{\pm 0.002}$ |
| + TDR-JL | $0.215_{\pm 0.003}$ | $0.728_{\pm 0.002}$ | $\underline{0.453}_{\pm 0.008}$ | $0.639_{\pm 0.010}$ | $0.247_{\pm 0.001}$ | $0.690_{\pm 0.002}$ | $0.422_{\pm 0.002}$ | $0.658_{\pm 0.003}$ |
| + StableDR | $\underline{0.209}_{\pm 0.002}$ | $0.737_{\pm 0.004}$ | $0.451_{\pm 0.008}$ | $0.640_{\pm 0.009}$ | $0.232_{\pm 0.001}$ | $0.689_{\pm 0.001}$ | $0.430_{\pm 0.003}$ | $0.659_{\pm 0.002}$ |
| + BEDR-JL | $\mathbf{0.207}_{\pm 0.002}$ | $\mathbf{0.759^*}_{\pm 0.005}$ | $\mathbf{0.466^*}_{\pm 0.006}$ | $\mathbf{0.663^*}_{\pm 0.007}$ | $\mathbf{0.198^*}_{\pm 0.002}$ | $\mathbf{0.692}_{\pm 0.002}$ | $\mathbf{0.436^*}_{\pm 0.005}$ | $\mathbf{0.664^*}_{\pm 0.003}$ |

rates several items to generate the 311,704 biased ratings. In addition, the first 5,400 users are asked to rate 10 randomly picked items, which constitutes the 54,000 unbiased ratings. We binarize the ratings to 0 for ratings less than 3, otherwise 1. In addition, we further use a fully exposed industrial dataset **KuaiRec** (Gao et al., 2022) with 4,676,570 video watching ratio records from 1,411 users to 3,327 videos. For this dataset, we binarize the records to 0 for records less than 2, otherwise 1.

**Baselines.** In our experiments, we take the probabilistic matrix factorization (PMF) (Mnih and Salakhutdinov, 2007) as the base model, and compare the proposed method with the information bottleneck based methods: **CVIB** (Wang et al., 2020) and **DIB** (Liu et al., 2021), the propensity based methods: **IPS** (Schnabel et al., 2016), **SNIPS** (Swaminathan and Joachims, 2015) and **ASIPS** (Saito, 2020), and the DR based methods: **DR** (Saito, 2020), **DR-JL** (Wang et al., 2019), **MRDR** (Guo et al., 2021), **DR-BIAS** (Dai et al., 2022), **DR-MSE** (Dai et al., 2022), **TDR** (Li et al., 2023b), **TDR-JL** (Li et al., 2023b), **StableDR** (Li et al., 2023c), and **MR** (Li et al., 2023a).

**Experimental Protocols and Details.** The following four metrics are used to measure the debiasing performance: MSE, AUC, Recall@K, NDCG@K, where we set K = 5 for **Coat** and **Yahoo! R3**, while set K = 20 for **KuaiRec**. All the experiments are implemented on PyTorch[3]. Adam is utilized as the optimizer for fast convergence in all experiments. We tune learning rate in $\{0.001, 0.005, 0.01, 0.05, 0.1\}$, batch size in $\{128, 256, 512\}$ for **Coat** and $\{1024, 2048, 4096\}$ for **Yahoo! R3** and **KuaiRec**. In addition, a standard normal distribution is adopted for the prior distribution of the user and item latent vector. Meanwhile, for the conditional distribution over the observed ratings, we use normal distribution with mean zero and variance $\sigma^2$, where $\sigma^2$ is tuned in $\{1e-5, 5e-5, 1e-4, 5e-4, 1e-3, 5e-3, 1e-2\}$.

## 5.2 Performance Comparison

Table 1 summarizes the debiasing performance of various methods on two real-world benchmark datasets **Coat** and **Yahoo! R3**. First, most debiased methods outperform the base model PMF, which shows the necessity for debiasing. Second, overall speaking, the information bottleneck based methods perform slightly better than the propensity based methods, while DR based methods such as TDR and StableDR demonstrate the most competitive performance. In addition, the proposed BEDR method achieves the best performance in terms of all the evaluation metrics. This is attributed to the proposed imputation balancing correction is able to effectively reduce the empirical covariance, which leads to a more unbiased estimation of the ideal loss function and results in significant improvement of debiasing performance. It empirically validates the effectiveness of the proposed method.

Table 3 reports the performance of various methods on a large scale industrial dataset **KuaiRec**. we can find that compared to baseline method PMF, vanilla DR method obtains little improvement,

---

[3]For all experiments, we use the Tesla T4 GPU as the computational resource

Table 2: Ablation study on **Coat** and **Yahoo! R3** datasets with mean and standard deviation of the empirical covariance (EC), AUC and NDCG@5. The best results are bolded, and $*$ means statistically significant results (p-value $\leq 0.01$) using the paired-t-test compared with the best baseline.

| | Ablation | | Coat | | | Yahoo! R3 | | |
|---|---|---|---|---|---|---|---|---|
| Method | Joint learning | EC control | EC $\downarrow$ | AUC $\uparrow$ | NDCG@5 $\uparrow$ | EC $\downarrow$ | AUC $\uparrow$ | NDCG@5 $\uparrow$ |
| DR | No | No | $2.135_{\pm0.127}$ | $0.728_{\pm0.013}$ | $0.636_{\pm0.019}$ | $8.249_{\pm0.426}$ | $0.686_{\pm0.003}$ | $0.657_{\pm0.002}$ |
| DR-JL | Yes | No | $1.567_{\pm0.109}$ | $0.735_{\pm0.008}$ | $0.630_{\pm0.004}$ | $5.688_{\pm0.295}$ | $0.686_{\pm0.002}$ | $0.650_{\pm0.003}$ |
| TDR | No | Partial | $1.415_{\pm0.072}$ | $0.719_{\pm0.006}$ | $0.630_{\pm0.010}$ | $1.707_{\pm0.078}$ | $0.687_{\pm0.002}$ | $0.661_{\pm0.002}$ |
| TDR-JL | Yes | Partial | $1.269_{\pm0.063}$ | $0.728_{\pm0.002}$ | $0.639_{\pm0.010}$ | $1.075_{\pm0.052}$ | $0.690_{\pm0.002}$ | $0.658_{\pm0.003}$ |
| BEDR-R | No | Regularizer | $2.048_{\pm0.116}$ | $0.728_{\pm0.006}$ | $0.633_{\pm0.007}$ | $5.379_{\pm0.272}$ | $0.686_{\pm0.002}$ | $0.658_{\pm0.003}$ |
| BEDR-JL-R | Yes | Regularizer | $1.468_{\pm0.105}$ | $0.736_{\pm0.004}$ | $0.634_{\pm0.006}$ | $4.681_{\pm0.231}$ | $0.687_{\pm0.002}$ | $0.656_{\pm0.002}$ |
| BEDR | No | Boosting | $0.388_{\pm0.085}$ | $0.738_{\pm0.004}$ | $0.645_{\pm0.006}$ | $0.584_{\pm0.083}$ | $0.690_{\pm0.002}$ | $0.662_{\pm0.002}$ |
| BEDR-JL | Yes | Boosting | $\mathbf{0.317}_{\pm0.077}$ | $\mathbf{0.759}^*_{\pm0.005}$ | $\mathbf{0.663}^*_{\pm0.007}$ | $\mathbf{0.502}_{\pm0.074}$ | $\mathbf{0.692}^*_{\pm0.002}$ | $\mathbf{0.664}^*_{\pm0.003}$ |

while DR-JL method achieves more performance improvement by jointly training the prediction model and the imputation model. The above two methods do not take the empirical covariance into account, which leads to a bias estimation of the ideal loss function and thus harm the prediction performance. Based on DR-JL method, our proposed BEDR-JL method modifies the imputation model with a subtle correction term and achieves substantial improvement on debiasing performance. This is because controlling the empirical covariance with the correction term provides a more accurate imputed errors, which can greatly boost the prediction performance. It further validates the simplicity and effectiveness of our proposed BEDR method.

Table 3: Performance comparison on **KuaiRec** dataset.

| | KuaiRec | | | |
|---|---|---|---|---|
| Method | MSE $\downarrow$ | AUC $\uparrow$ | Recall@20 $\uparrow$ | NDCG@20 $\uparrow$ |
| PMF | $0.137_{\pm0.000}$ | $0.754_{\pm0.001}$ | $0.360_{\pm0.001}$ | $0.480_{\pm0.002}$ |
| + IPS | $0.050_{\pm0.002}$ | $0.746_{\pm0.008}$ | $0.389_{\pm0.008}$ | $0.459_{\pm0.007}$ |
| + SNIPS | $0.047_{\pm0.001}$ | $0.751_{\pm0.002}$ | $0.368_{\pm0.003}$ | $0.447_{\pm0.002}$ |
| + ASIPS | $0.096_{\pm0.001}$ | $0.751_{\pm0.007}$ | $0.393_{\pm0.012}$ | $0.477_{\pm0.007}$ |
| + DR | $0.046_{\pm0.001}$ | $0.744_{\pm0.006}$ | $0.383_{\pm0.011}$ | $0.480_{\pm0.011}$ |
| + DR-JL | $0.046_{\pm0.001}$ | $0.762_{\pm0.004}$ | $0.393_{\pm0.006}$ | $0.463_{\pm0.009}$ |
| + CVIB | $0.084_{\pm0.001}$ | $0.765_{\pm0.001}$ | $0.406_{\pm0.004}$ | $0.453_{\pm0.006}$ |
| + MRDR | $0.046_{\pm0.001}$ | $0.767_{\pm0.004}$ | $0.403_{\pm0.006}$ | $0.472_{\pm0.008}$ |
| + DIB | $\underline{0.045}_{\pm0.000}$ | $0.768_{\pm0.001}$ | $\underline{0.411}_{\pm0.001}$ | $0.480_{\pm0.001}$ |
| + DR-BIAS | $0.045_{\pm0.001}$ | $0.765_{\pm0.005}$ | $0.398_{\pm0.007}$ | $0.468_{\pm0.011}$ |
| + DR-MSE | $0.046_{\pm0.001}$ | $0.768_{\pm0.002}$ | $0.403_{\pm0.009}$ | $0.472_{\pm0.007}$ |
| + MR | $0.118_{\pm0.001}$ | $0.769_{\pm0.002}$ | $0.388_{\pm0.007}$ | $0.483_{\pm0.007}$ |
| + TDR | $0.134_{\pm0.001}$ | $0.767_{\pm0.002}$ | $0.383_{\pm0.003}$ | $0.488_{\pm0.003}$ |
| + TDR-JL | $0.123_{\pm0.001}$ | $0.770_{\pm0.004}$ | $0.395_{\pm0.003}$ | $0.490_{\pm0.002}$ |
| + StableDR | $0.115_{\pm0.001}$ | $\underline{0.773}_{\pm0.002}$ | $0.389_{\pm0.002}$ | $\underline{0.489}_{\pm0.002}$ |
| + BEDR-JL | $0.048_{\pm0.002}$ | $\mathbf{0.779}^*_{\pm0.002}$ | $\mathbf{0.423}^*_{\pm0.006}$ | $\mathbf{0.496}^*_{\pm0.006}$ |

Table 4: Ablation study on **KuaiRec** dataset.

| | KuaiRec | | |
|---|---|---|---|
| Method | EC $\downarrow$ | AUC $\uparrow$ | NDCG@20 $\uparrow$ |
| DR | $8.042_{\pm0.467}$ | $0.744_{\pm0.006}$ | $0.480_{\pm0.011}$ |
| DR-JL | $2.324_{\pm0.133}$ | $0.762_{\pm0.004}$ | $0.463_{\pm0.009}$ |
| TDR | $1.802_{\pm0.113}$ | $0.767_{\pm0.002}$ | $0.488_{\pm0.003}$ |
| TDR-JL | $1.028_{\pm0.088}$ | $0.770_{\pm0.004}$ | $0.490_{\pm0.002}$ |
| BEDR-R | $2.108_{\pm0.098}$ | $0.763_{\pm0.003}$ | $0.466_{\pm0.006}$ |
| BEDR-JL-R | $1.524_{\pm0.098}$ | $0.772_{\pm0.002}$ | $0.484_{\pm0.005}$ |
| BEDR | $1.111_{\pm0.092}$ | $0.776_{\pm0.002}$ | $0.492_{\pm0.004}$ |
| BEDR-JL | $\mathbf{0.955}^*_{\pm0.088}$ | $\mathbf{0.779}^*_{\pm0.002}$ | $\mathbf{0.496}^*_{\pm0.006}$ |

## 5.3 ABLATION STUDY

As mentioned in Section 4.2 and 4.4, the empirical covariance plays a crucial role in both unbiased estimation and unbiased learning. To explore the relationship between the degree of empirical covariance control and the performance, we conduct ablation study on all three datasets with empirical covariance (EC), AUC, and NDCG@K as the evaluation metrics. We compare the proposed BEDR method with DR and TDR methods, and we denote the method that directly regard the observed empirical covariance as a regularization term as BEDR-R and BEDR-JL-R, respectively. DR method has no control on empirical covariance at all, while TDR method can control the observed empirical covariance by adding $o_{u,i}\left(\frac{1}{\hat{p}_{u,i}} - 1\right)$ as the correction term to the imputed errors. We fix a pretrained imputation model for the methods without learning. The results are shown in Table 2 and Table 4.

**Effects of Empirical Covariance Control.** Compared to the baseline methods, methods that are able to control empirical covariance have improved debiasing performance, which indicates the necessity to control empirical covariance for debiasing. In addition, TDR method obtains some performance improvement because it guarantees the control of empirical covariance for the observed outcomes. Howerver, TDR method cannot control the empirical covariance for missing outcomes. Furthermore, the proposed BEDR method achieves the most significant empirical covariance decreases and the most competitive performance in AUC and NDCG@K. This empirically demonstrates that the reduction of

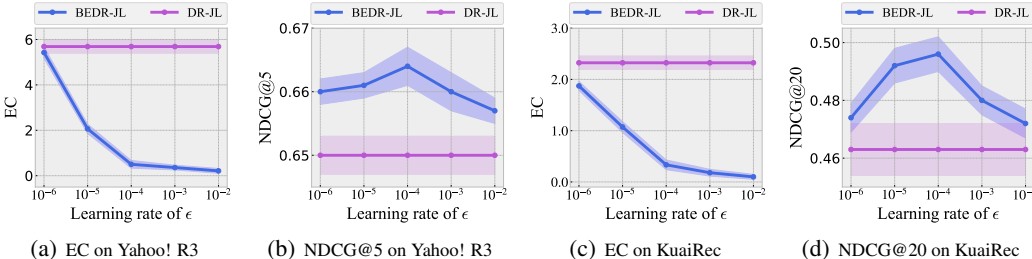

Figure 1: Effects of learning rate of correction hyperparameter $\epsilon$ on EC and NDCG@K.

empirical covariance contributes to the prediction performance. Moreover, directly regard empirical covariance as a regularizer fails to achieve the desired performance and is beaten by TDR method. On one hand, it can only control the observed empirical covariance. On the other hand, it cannot guarantee the observed empirical covariance to be exact zero.

**Effects of Joint Learning.** From Table 2 and Table 4, almost all methods with joint learning can reduce the empirical covariance value and improve the debiasing performance, which is attributed to joint learning can bridge the prediction model and the imputation model effectively to improve the accuracy of predicted ratings and imputed errors. Overall, BEDR-JL method exhibits the optimal performance on all three datasets. Nevertheless, the improvement on performance will be limited if only apply the joint learning technique without controlling the empirical covariance. This further illustrates the necessity of empirical covariance control.

## 5.4 SENSITIVITY ANALYSIS

As discussed in Section 4.2, the proposed BEDR method modifies the imputation model to control the empirical covariance with a learnable parameter $\epsilon$. Therefore, it is meaningful to explore the relationship bewteen the learning rate of $\epsilon$ and the debiasing performance. Figure 1 shows the AUC and NDCG@K performance under varying learning rate of $\epsilon$ on **Yahoo! R3** and **KuaiRec**, respectively. On one hand, as the learning rate increases, the empirical covariance value of the proposed BEDR-JL method monotonically decreases and outperforms DR-JL, which demonstrates the effectiveness of the proposed imputation balancing correction. On the other hand, it should be noted that due to the no harm property, the bias reduction property and the variance reduction property in Section 4.2, the proposed method outperforms DR-JL under varying learning rate. Finally, the proposed method exhibits the optimal performance at a relatively moderate learning rate about 1e-4.

## 6 CONCLUSION

By noting that previous studies on debiasing have relied on the deterministic models for estimating the prediction errors and propensities, i.e., only rating missing indicators are treated as random variables. In this paper, we theoretically relax the requirement of the deterministic models and consider the case of using probabilistic models, i.e., all variables including ratings, propensity and imputation errors are random. When using the probabilistic models for estimating the prediction errors and propensities, we demonstrate theoretically that controlling empirical covariates helps reduce bias. We then propose a novel balancing enhanced doubly robust joint learning method to achieve unbiasedness. The proposed method makes a subtle change based on the previous DR method by adding a balancing correction term to the imputation model. Theoretical analysis shows that the proposed method can lead to significant performance improvement by using the imputation balancing correction to obtain the high quality imputed errors. In addition, extensive experiments are conducted on three real-world datasets, including a large-scale industrial dataset, which verifies the effectiveness of the proposed method for debiased recommendation. This article opens a new perspective on the theory for debiasing recommendations under probabilistic models, and the proposed approach is simple and effective. A possible shortcoming is that despite the favorable empirical performance shown by joint training, theoretical guarantees are still lacking. We leave the theoretical guarantees of training under probabilistic models as a generalization of unbiased estimation for future research.

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

# A PROOFS

**Lemma 1** (Bias of DR Estimator under **Deterministic Models** (Wang et al., 2019)). *Given imputed errors $\hat{e}_{u,i}$ and learned propensities $\hat{p}_{u,i} > 0$ for all user-item pairs, when considering only the randomness of rating missing indicators, the bias of the DR estimator is*

$$\text{Bias}_{\mathcal{O}}[\mathcal{E}_{DR}(\theta)] = \frac{1}{|\mathcal{D}|} \sum_{(u,i)\in\mathcal{D}} \frac{\{\hat{p}_{u,i} - p_{u,i}\} \cdot \{e_{u,i} - \hat{e}_{u,i}\}}{\hat{p}_{u,i}}.$$

*Proof of Lemma 1.* The proof can be found in Lemma 3.1 of Wang et al. (2019). However, one should note that, as stated in the proof, "the prediction and imputed errors are treated as constants when taking the expectation, since $o_{u,i}$ does not result from any prediction or imputation models (Schnabel et al., 2016)". The DR estimator in (Wang et al., 2019) is given as

$$\mathcal{E}_{DR}(\theta) = \frac{1}{|\mathcal{D}|} \sum_{(u,i)\in\mathcal{D}} \left[ \hat{e}_{u,i} + \frac{o_{u,i}(e_{u,i} - \hat{e}_{u,i})}{\hat{p}_{u,i}} \right].$$

By considering only the randomness on $o_{u,i}$, we have

$$\mathbb{E}_{\mathcal{O}}[\mathcal{E}_{DR}(\theta)] = \mathbb{E}_{\mathcal{O}}\left[ \frac{1}{|\mathcal{D}|} \sum_{(u,i)\in\mathcal{D}} \left[ \hat{e}_{u,i} + \frac{o_{u,i}(e_{u,i} - \hat{e}_{u,i})}{\hat{p}_{u,i}} \right] \right]$$

$$= \frac{1}{|\mathcal{D}|} \sum_{(u,i)\in\mathcal{D}} \left[ \hat{e}_{u,i} + \frac{p_{u,i}(e_{u,i} - \hat{e}_{u,i})}{\hat{p}_{u,i}} \right].$$

By definition, the bias of the DR estimator is

$$\text{Bias}_{\mathcal{O}}[\mathcal{E}_{DR}(\theta)] = \mathcal{E}_{ideal}(\theta) - \mathbb{E}_{\mathcal{O}}[\mathcal{E}_{DR}(\theta)]$$

$$= \frac{1}{|\mathcal{D}|} \sum_{(u,i)\in\mathcal{D}} e_{u,i} - \frac{1}{|\mathcal{D}|} \sum_{(u,i)\in\mathcal{D}} \left[ \hat{e}_{u,i} + \frac{p_{u,i}(e_{u,i} - \hat{e}_{u,i})}{\hat{p}_{u,i}} \right]$$

$$= \frac{1}{|\mathcal{D}|} \sum_{(u,i)\in\mathcal{D}} \frac{\{\hat{p}_{u,i} - p_{u,i}\} \cdot \{e_{u,i} - \hat{e}_{u,i}\}}{\hat{p}_{u,i}},$$

which yields the stated results. $\square$

**Corollary 1** (Double Robustness under **Deterministic Models** (Wang et al., 2019)). *The DR estimator is unbiased when either imputed errors $\hat{e}_{u,i}$ or learned propensities $\hat{p}_{u,i}$ are accurate for all user-item pairs, i.e., either $\hat{e}_{u,i} = e_{u,i}$ or $\hat{p}_{u,i} = p_{u,i}$.*

*Proof of Corollary 1.* The proof can be found at Corollary 3.1 in Appendix of (Wang et al., 2019). However, one should note that, as stated in the proof, "the prediction and imputed errors are treated as constants when taking the expectation, since $o_{u,i}$ does not result from any prediction or imputation models (Schnabel et al., 2016)".

Let $\delta_{u,i} = e_{u,i} - \hat{e}_{u,i}$ and $\Delta_{u,i} = \frac{\hat{p}_{u,i} - p_{u,i}}{\hat{p}_{u,i}}$. On the hand, when imputed errors are accurate, we have $\delta_{u,i} = 0$ for $(u,i) \in \mathcal{D}$. In such case, we can compute the bias of the DR estimator by

$$\text{Bias}_{\mathcal{O}}[\mathcal{E}_{DR}(\theta)] = \frac{1}{|\mathcal{D}|} \sum_{u,i\in\mathcal{D}} \Delta_{u,i}\delta_{u,i} = \frac{1}{|\mathcal{D}|} \sum_{u,i\in\mathcal{D}} \Delta_{u,i} \cdot 0 = 0.$$

On the other hand, when the learned propensities are accurate, we have $\Delta_{u,i} = 0$ for $(u,i) \in \mathcal{D}$. In this case, we can compute the bias of the DR estimator by

$$\text{Bias}(\mathcal{E}_{\text{DR}}) = \frac{1}{|\mathcal{D}|} \sum_{u,i\in\mathcal{D}} \Delta_{u,i}\delta_{u,i} = \frac{1}{|\mathcal{D}|} \sum_{u,i\in\mathcal{D}} 0 \cdot \delta_{u,i} = 0.$$

In both cases, the bias of the DR estimator is zero, which means that the expectation of the DR estimator over all the possible instances of $o_{u,i}$ is exactly the same as the prediction inaccuracy. This completes the proof. $\square$

**Theorem 1** (Bias of DR Estimator under **Probabilistic Models**). *Given probabilistic error imputation model* $\mathbb{P}(\hat{e}_{u,i} \mid x_{u,i})$ *and probabilistic propensity model* $\mathbb{P}(\hat{p}_{u,i} \mid x_{u,i})$, *consider all variables are random, then the bias of the DR estimator, namely* $\mathrm{Bias}[\mathcal{E}_{DR}(\theta)]$, *is*

$$\underbrace{\mathrm{Cov}\left(\frac{\hat{p}_{u,i} - o_{u,i}}{\hat{p}_{u,i}}, e_{u,i} - \hat{e}_{u,i}\right)}_{\textit{equals to 0 if independent}} + \underbrace{\mathbb{E}\left[\left\{1 - \mathbb{E}\left[\frac{o_{u,i}}{\hat{p}_{u,i}}\Big|x_{u,i}\right]\right\} \cdot \{\mathbb{E}[e_{u,i} \mid x_{u,i}] - \mathbb{E}[\hat{e}_{u,i} \mid x_{u,i}]\}\right]}_{\textit{equals to 0 either } \mathbb{E}[o_{u,i}/\hat{p}_{u,i} \mid x_{u,i}] = 1 \textit{ or } \mathbb{E}[\hat{e}_{u,i} - e_{u,i} \mid x_{u,i}] = 0}$$

*Proof of Theorem 1.* Instead of considering only the randomness of the rating missing indicator, in the following, we treat all variables, including imputed errors and learned propensities, as random variables. Formally, we have

$$\mathrm{Bias}[\mathcal{E}_{DR}(\theta)] = \mathbb{E}[\mathcal{E}_{ideal}(\theta)] - \mathbb{E}[\mathcal{E}_{DR}(\theta)] = \mathbb{E}[e_{u,i}] - \mathbb{E}\left[e_{u,i} + \frac{\{o_{u,i} - \hat{p}_{u,i}\} \cdot \{e_{u,i} - \hat{e}_{u,i}\}}{\hat{p}_{u,i}}\right]$$

$$= \mathbb{E}\left[\mathbb{E}\left[\left\{\frac{\hat{p}_{u,i} - o_{u,i}}{\hat{p}_{u,i}}\right\}\{e_{u,i} - \hat{e}_{u,i}\} \mid x_{u,i}\right]\right] \qquad \text{(by the double expectation formula)}$$

$$= \mathbb{E}\left[\mathbb{E}\left[\left\{\frac{\hat{p}_{u,i} - o_{u,i}}{\hat{p}_{u,i}} - \mathbb{E}\left[\frac{\hat{p}_{u,i} - o_{u,i}}{\hat{p}_{u,i}}\right] + \mathbb{E}\left[\frac{\hat{p}_{u,i} - o_{u,i}}{\hat{p}_{u,i}}\right]\right\}\{(e_{u,i} - \hat{e}_{u,i}) - \mathbb{E}[e_{u,i} - \hat{e}_{u,i}] + \mathbb{E}[e_{u,i} - \hat{e}_{u,i}]\} \mid x_{u,i}\right]\right]$$

$$= \mathbb{E}\left[\mathbb{E}\left[\left\{\frac{\hat{p}_{u,i} - o_{u,i}}{\hat{p}_{u,i}} - \mathbb{E}\left[\frac{\hat{p}_{u,i} - o_{u,i}}{\hat{p}_{u,i}}\right]\right\}\{(e_{u,i} - \hat{e}_{u,i}) - \mathbb{E}[e_{u,i} - \hat{e}_{u,i}]\} \mid x_{u,i}\right]\right]$$

$$+ \mathbb{E}\left[\left\{1 - \mathbb{E}\left[\frac{o_{u,i}}{\hat{p}_{u,i}}\Big|x_{u,i}\right]\right\} \cdot \{\mathbb{E}[e_{u,i} \mid x_{u,i}] - \mathbb{E}[\hat{e}_{u,i} \mid x_{u,i}]\}\right]$$

$$= \mathrm{Cov}\left(\frac{\hat{p}_{u,i} - o_{u,i}}{\hat{p}_{u,i}}, e_{u,i} - \hat{e}_{u,i}\right) + \mathbb{E}\left[\left\{1 - \mathbb{E}\left[\frac{o_{u,i}}{\hat{p}_{u,i}}\Big|x_{u,i}\right]\right\} \cdot \{\mathbb{E}[e_{u,i} \mid x_{u,i}] - \mathbb{E}[\hat{e}_{u,i} \mid x_{u,i}]\}\right],$$

which yields the stated results. $\qquad\square$

**Corollary 2** (Double Robustness under **Probabilistic Models**). *The DR estimator is unbiased when both the following conditions hold:*

*(i) conditional independence condition holds, i.e.,* $\mathrm{Cov}((\hat{p}_{u,i} - o_{u,i})/\hat{p}_{u,i}, e_{u,i} - \hat{e}_{u,i}) = 0$;

*(ii) either learned propensities satisfy* $\mathbb{E}[o_{u,i}/\hat{p}_{u,i} \mid x_{u,i}] = 1$, *or imputed errors have the same conditional expectation with true prediction errors* $\mathbb{E}[\hat{e}_{u,i} \mid x_{u,i}] = \mathbb{E}[e_{u,i} \mid x_{u,i}]$.

*Proof of Corollary 2.* First, when condition (i) holds, i.e.,

$$\mathrm{Cov}((\hat{p}_{u,i} - o_{u,i})/\hat{p}_{u,i}, e_{u,i} - \hat{e}_{u,i}) = 0,$$

it follows from the results in Theorem 1 that

$$\mathrm{Bias}[\mathcal{E}_{DR}(\theta)] = \mathbb{E}\left[\left\{1 - \mathbb{E}\left[\frac{o_{u,i}}{\hat{p}_{u,i}}\Big|x_{u,i}\right]\right\} \cdot \{\mathbb{E}[e_{u,i} \mid x_{u,i}] - \mathbb{E}[\hat{e}_{u,i} \mid x_{u,i}]\}\right]$$

On the hand, when the learned propensities satisfy $\mathbb{E}[o_{u,i}/\hat{p}_{u,i} \mid x_{u,i}] = 1$. In such case, we can compute the bias of the DR estimator by

$$\mathrm{Bias}[\mathcal{E}_{DR}(\theta)] = \mathbb{E}\left[0 \cdot \{\mathbb{E}[e_{u,i} \mid x_{u,i}] - \mathbb{E}[\hat{e}_{u,i} \mid x_{u,i}]\}\right] = 0.$$

On the other hand, when imputed errors have the same conditional expectation with true prediction errors, we have $\mathbb{E}[\hat{e}_{u,i} \mid x_{u,i}] = \mathbb{E}[e_{u,i} \mid x_{u,i}]$. In this case, we can compute the bias of the DR estimator by

$$\mathrm{Bias}[\mathcal{E}_{DR}(\theta)] = \mathbb{E}\left[\left\{1 - \mathbb{E}\left[\frac{o_{u,i}}{\hat{p}_{u,i}}\Big|x_{u,i}\right]\right\} \cdot 0\right] = 0.$$

In both cases, the bias of the DR estimator is zero, which completes the proof. $\qquad\square$

**Definition 1** (Empirical Covariance). *The empirical expected conditional covariance between $(\hat{p}_{u,i} - o_{u,i})/\hat{p}_{u,i}$ and $e_{u,i} - \hat{e}_{u,i}$ is*

$$\widehat{\text{Cov}}\left(\frac{\hat{p}_{u,i} - o_{u,i}}{\hat{p}_{u,i}}, e_{u,i} - \hat{e}_{u,i}\right) = \frac{1}{|\mathcal{D}|}\sum_{(u,i)\in\mathcal{D}}\frac{\hat{p}_{u,i} - o_{u,i}}{\hat{p}_{u,i}} \cdot (e_{u,i} - \hat{e}_{u,i}).$$

**Definition 2** (Empirical Rademacher Complexity (Shalev-Shwartz and Ben-David, 2014)). *Let $\mathcal{F}$ be a family of prediction models mapping from $x \in \mathcal{X}$ to $[a, b]$, and $S = \{x_{u,i} \mid (u,i) \in \mathcal{D}\}$ a fixed sample of size $|\mathcal{D}|$ with elements in $\mathcal{X}$. Then, the empirical Rademacher complexity of $\mathcal{F}$ with respect to the sample $S$ is defined as:*

$$\mathcal{R}(\mathcal{F}) = \mathbb{E}_{\boldsymbol{\sigma}\sim\{-1,+1\}^{|\mathcal{D}|}}\sup_{f_\theta\in\mathcal{F}}\left[\frac{1}{|\mathcal{D}|}\sum_{(u,i)\in\mathcal{D}}\sigma_{u,i}e_{u,i}\right],$$

*where $\boldsymbol{\sigma} = \{\sigma_{u,i} : (u,i) \in \mathcal{D}\}$, and $\sigma_{u,i}$ are independent uniform random variables taking values in $\{-1, +1\}$. The random variables $\sigma_{u,i}$ are called Rademacher variables.*

**Lemma 2** (Rademacher Comparison Lemma (Shalev-Shwartz and Ben-David, 2014)). *Let $\mathcal{F}$ be a family of real-valued functions on $z \in \mathcal{Z}$ to $[a, b]$, and $S = \{x_{u,i} \mid (u,i) \in \mathcal{D}\}$ a fixed sample of size $|\mathcal{D}|$ with elements in $\mathcal{X}$. Then*

$$\mathbb{E}_{S\sim\mathbb{P}^{|\mathcal{D}|}}\left[\sup_{f\in\mathcal{F}}\left(\mathbb{E}_{z\sim\mathbb{P}}[f(z)] - \frac{1}{|\mathcal{D}|}\sum_{(u,i)\in\mathcal{D}}f(z_{u,i})\right)\right] \leq 2\mathbb{E}_{S\sim\mathbb{P}^{|\mathcal{D}|}}\mathbb{E}_{\boldsymbol{\sigma}\sim\{-1,+1\}^{|\mathcal{D}|}}\sup_{f\in\mathcal{F}}\left[\frac{1}{|\mathcal{D}|}\sum_{(u,i)\in\mathcal{D}}\sigma_{u,i}f(z_{u,i})\right],$$

*where $\boldsymbol{\sigma} = \{\sigma_{u,i} : (u,i) \in \mathcal{D}\}$, and $\sigma_{u,i}$ are independent uniform random variables taking values in $\{-1, +1\}$. The random variables $\sigma_{u,i}$ are called Rademacher variables.*

*Proof of Lemma 2.* The proof can be found in Lemma 26.2 of (Shalev-Shwartz and Ben-David, 2014). $\square$

**Lemma 3** (McDiarmid's Inequality (Shalev-Shwartz and Ben-David, 2014)). *Let $V$ be some set and let $f : V^m \to \mathbb{R}$ be a function of $m$ variables such that for some $c > 0$, for all $i \in [m]$ and for all $x_1, \ldots, x_m, x_i' \in V$ we have*

$$|f(x_1, \ldots, x_m) - f(x_1, \ldots, x_{i-1}, x_i', x_{i+1}, \ldots, x_m)| \leq c$$

*Let $X_1, \ldots, X_m$ be $m$ independent random variables taking values in $V$. Then, with probability of at least $1 - \delta$ we have*

$$|f(X_1, \ldots, X_m) - \mathbb{E}[f(X_1, \ldots, X_m)]| \leq c\sqrt{\log\left(\frac{2}{\delta}\right)m/2}$$

*Proof of Lemma 3.* The proof can be found in Lemma 26.4 of (Shalev-Shwartz and Ben-David, 2014). $\square$

**Lemma 4** (Rademacher Calculus (Shalev-Shwartz and Ben-David, 2014)). *For any $A \subset \mathbb{R}^m$, scalar $c \in \mathbb{R}$, and vector $\mathbf{a}_0 \in \mathbb{R}^m$, we have*

$$R(\{c\mathbf{a} + \mathbf{a}_0 : \mathbf{a} \in A\}) \leq |c|R(A).$$

*Proof of Lemma 4.* The proof can be found in Lemma 26.6 of (Shalev-Shwartz and Ben-David, 2014). $\square$

**Theorem 2** (Controllability of Empirical Covariance). *The boosted imputation model trained by minimizing the balanced enhanced imputation loss is sufficient for controlling the empirical covariance.*

*(i) For user-item pairs with **observed** outcomes, the empirical covariance is 0. Formally, we have*

$$\frac{\partial}{\partial \epsilon} \mathcal{L}_e^{Bal}(\phi, \epsilon) \Big|_{\epsilon=\epsilon^*} = 0, \quad \text{which is equivalent to} \quad \frac{1}{|\mathcal{D}|} \sum_{(u,i):\, o_{u,i}=1} \frac{\hat{p}_{u,i} - o_{u,i}}{\hat{p}_{u,i}} \cdot (e_{u,i} - \tilde{e}_{u,i}) = 0;$$

*(ii) For user-item pairs with **missing** outcomes, suppose that $\hat{p}_{u,i} \geq K_\psi$ and $|e_{u,i} - \hat{e}_{u,i}| \leq K_\phi$, then with probability at least $1 - \eta$, we have*

$$\frac{1}{|\mathcal{D}|} \sum_{(u,i):\, o_{u,i}=0} \frac{\hat{p}_{u,i} - o_{u,i}}{\hat{p}_{u,i}} \cdot (e_{u,i} - \tilde{e}_{u,i}) \leq \underbrace{\mathcal{L}_e^{Bal}(\phi, \epsilon)^{\frac{1}{2}}}_{proposed\ loss} + K_\phi \cdot \underbrace{\left[ \frac{1}{|\mathcal{D}|} \sum_{u,i \in \mathcal{D}} \left| 1 - \mathbb{E}\left[ \frac{o_{u,i}}{\hat{p}_{u,i}} \Big| x_{u,i} \right] \right| \right]^{\frac{1}{2}}}_{empirical\ bias\ from\ probabilistic\ propensity\ model}$$

$$+ \underbrace{\left[ K_\phi \left( 1 + \frac{1}{K_\psi} \right) \left( 2\mathcal{R}(\mathcal{F}) + K_\phi \sqrt{\frac{18 \log(4/\eta)}{|\mathcal{D}|}} \right) \right]^{\frac{1}{2}}}_{tail\ bound\ controlled\ by\ empirical\ Rademacher\ complexity\ and\ sample\ size}$$

*Proof.* For the proof of Theorem 2(i), first recap that the proposed boosted imputation model is

$$\tilde{e}_{u,i} = m(x_{u,i}; \phi) + \epsilon(o_{u,i} - \pi(x_{u,i}; \psi)),$$

and the proposed balancing enhanced loss function for training the boosted imputation model is

$$(\phi^*, \epsilon^*) = \arg\min_{\phi, \epsilon} \mathcal{L}_e^{Bal}(\phi, \epsilon) = \frac{1}{|\mathcal{D}|} \sum_{(u,i) \in \mathcal{D}} \frac{o_{u,i}(e_{u,i} - \tilde{e}_{u,i})^2}{\hat{p}_{u,i}}.$$

By taking the partial derivative with respective to $\epsilon$ of the above formula and setting it to zero, we have

$$\frac{\partial}{\partial \epsilon} \mathcal{L}_e^{Bal}(\phi, \epsilon) \Big|_{\epsilon=\epsilon^*} = 0, \quad \text{which is equivalent to} \quad \frac{1}{|\mathcal{D}|} \sum_{(u,i):\, o_{u,i}=1} \frac{\hat{p}_{u,i} - o_{u,i}}{\hat{p}_{u,i}} \cdot (e_{u,i} - \tilde{e}_{u,i}) = 0,$$

which proves the empirical convariance on the observed outcomes is 0.

For the proof of Theorem 2(ii), by noting that

$$\frac{1}{|\mathcal{D}|} \sum_{(u,i):\, o_{u,i}=0} \frac{\hat{p}_{u,i} - o_{u,i}}{\hat{p}_{u,i}} \cdot (e_{u,i} - \tilde{e}_{u,i}) = \frac{1}{|\mathcal{D}|} \sum_{(u,i):\, o_{u,i}=0} (e_{u,i} - \tilde{e}_{u,i}) \leq \left[ \frac{1}{|\mathcal{D}|} \sum_{(u,i) \in \mathcal{D}} (e_{u,i} - \tilde{e}_{u,i})^2 \right]^{\frac{1}{2}},$$

we now focus on bounding the last term of the above equation with the least probability.

Suppose that $\hat{p}_{u,i} \geq K_\psi$ and $|e_{u,i} - \tilde{e}_{u,i}| \leq K_\phi$, then

$$\frac{1}{|\mathcal{D}|} \sum_{(u,i)\in\mathcal{D}} (e_{u,i} - \tilde{e}_{u,i})^2 = \frac{1}{|\mathcal{D}|} \sum_{(u,i)\in\mathcal{D}} \frac{o_{u,i}(e_{u,i} - \tilde{e}_{u,i})^2}{\hat{p}_{u,i}} + \frac{1}{|\mathcal{D}|} \sum_{(u,i)\in\mathcal{D}} (e_{u,i} - \tilde{e}_{u,i})^2$$

$$- \mathbb{E}\left[\frac{1}{|\mathcal{D}|} \sum_{(u,i)\in\mathcal{D}} \frac{o_{u,i}(e_{u,i} - \tilde{e}_{u,i})^2}{\hat{p}_{u,i}}\right] + \mathbb{E}\left[\frac{1}{|\mathcal{D}|} \sum_{(u,i)\in\mathcal{D}} \frac{o_{u,i}(e_{u,i} - \tilde{e}_{u,i})^2}{\hat{p}_{u,i}}\right] - \frac{1}{|\mathcal{D}|} \sum_{(u,i)\in\mathcal{D}} \frac{o_{u,i}(e_{u,i} - \tilde{e}_{u,i})^2}{\hat{p}_{u,i}}$$

$$\leq \frac{1}{|\mathcal{D}|} \sum_{(u,i)\in\mathcal{D}} \frac{o_{u,i}(e_{u,i} - \tilde{e}_{u,i})^2}{\hat{p}_{u,i}} + \left|\frac{1}{|\mathcal{D}|} \sum_{(u,i)\in\mathcal{D}} (e_{u,i} - \tilde{e}_{u,i})^2 - \mathbb{E}\left[\frac{1}{|\mathcal{D}|} \sum_{(u,i)\in\mathcal{D}} \frac{o_{u,i}(e_{u,i} - \tilde{e}_{u,i})^2}{\hat{p}_{u,i}}\right]\right|$$

$$+ \left(\mathbb{E}\left[\frac{1}{|\mathcal{D}|} \sum_{(u,i)\in\mathcal{D}} \frac{o_{u,i}(e_{u,i} - \tilde{e}_{u,i})^2}{\hat{p}_{u,i}}\right] - \frac{1}{|\mathcal{D}|} \sum_{(u,i)\in\mathcal{D}} \frac{o_{u,i}(e_{u,i} - \tilde{e}_{u,i})^2}{\hat{p}_{u,i}}\right)$$

$$\leq \mathcal{L}_e^{Bal}(\phi,\epsilon) + K_\phi^2 \cdot \left|\mathbb{E}\left[\frac{1}{|\mathcal{D}|} \sum_{(u,i)\in\mathcal{D}} 1 - \frac{o_{u,i}}{\hat{p}_{u,i}}\right]\right|$$

$$+ \sup_{f_\theta\in\mathcal{F}} \left(\mathbb{E}\left[\frac{1}{|\mathcal{D}|} \sum_{(u,i)\in\mathcal{D}} \frac{o_{u,i}(e_{u,i} - \tilde{e}_{u,i})^2}{\hat{p}_{u,i}}\right] - \frac{1}{|\mathcal{D}|} \sum_{(u,i)\in\mathcal{D}} \frac{o_{u,i}(e_{u,i} - \tilde{e}_{u,i})^2}{\hat{p}_{u,i}}\right).$$

For simplicity, we denote the last term in the above formula as

$$\mathcal{B}(\mathcal{F}) = \sup_{f_\theta\in\mathcal{F}} \left(\mathbb{E}\left[\frac{1}{|\mathcal{D}|} \sum_{(u,i)\in\mathcal{D}} \frac{o_{u,i}(e_{u,i} - \tilde{e}_{u,i})^2}{\hat{p}_{u,i}}\right] - \frac{1}{|\mathcal{D}|} \sum_{(u,i)\in\mathcal{D}} \frac{o_{u,i}(e_{u,i} - \tilde{e}_{u,i})^2}{\hat{p}_{u,i}}\right),$$

we then aim to bound $\mathcal{B}(\mathcal{F})$ in the following.

Note that

$$\mathcal{B}(\mathcal{F}) = \mathbb{E}_{S\sim\mathbb{P}^{|\mathcal{D}|}}[\mathcal{B}(\mathcal{F})] + \left\{\mathcal{B}(\mathcal{F}) - \mathbb{E}_{S\sim\mathbb{P}^{|\mathcal{D}|}}[\mathcal{B}(\mathcal{F})]\right\},$$

where the first term is $\mathbb{E}_{S\sim\mathbb{P}^{|\mathcal{D}|}}[\mathcal{B}(\mathcal{F})]$, and by Lemma 2 we have

$$\mathbb{E}_{S\sim\mathbb{P}^{|\mathcal{D}|}}[\mathcal{B}(\mathcal{F})] \leq 2 \mathbb{E}_{S\sim\mathbb{P}^{|\mathcal{D}|}} \mathbb{E}_{\boldsymbol{\sigma}\sim\{-1,+1\}^{|\mathcal{D}|}} \sup_{f_\theta\in\mathcal{F}} \left[\frac{1}{|\mathcal{D}|} \sum_{(u,i)\in\mathcal{D}} \sigma_{u,i} \frac{o_{u,i}(e_{u,i} - \tilde{e}_{u,i})^2}{\hat{p}_{u,i}}\right].$$

By the assumptions that $\hat{p}_{u,i} \geq K_\psi$ and $|e_{u,i} - \tilde{e}_{u,i}| \leq K_\phi$, we have

$$\mathbb{E}_{S\sim\mathbb{P}^{|\mathcal{D}|}}[\mathcal{B}(\mathcal{F})] \leq 2K_\phi \left(1 + \frac{1}{K_\psi}\right) \mathbb{E}_{S\sim\mathbb{P}^{|\mathcal{D}|}} \mathbb{E}_{\boldsymbol{\sigma}\sim\{-1,+1\}^{|\mathcal{D}|}} \sup_{f_\theta\in\mathcal{F}} \left[\frac{1}{|\mathcal{D}|} \sum_{(u,i)\in\mathcal{D}} \sigma_{u,i}(e_{u,i} - \tilde{e}_{u,i})\right]$$

$$= 2K_\phi \left(1 + \frac{1}{K_\psi}\right) \mathbb{E}_{S\sim\mathbb{P}^{|\mathcal{D}|}} \{\mathcal{R}(\mathcal{F})\},$$

where the last equation is directly from Lemma 4, and $\mathcal{R}(\mathcal{F})$ is the empirical Rademacher complexity

$$\mathcal{R}(\mathcal{F}) = \mathbb{E}_{\boldsymbol{\sigma}\sim\{-1,+1\}^{|\mathcal{D}|}} \sup_{f_\theta\in\mathcal{F}} \left[\frac{1}{|\mathcal{D}|} \sum_{(u,i)\in\mathcal{D}} \sigma_{u,i} e_{u,i}\right],$$

where $\boldsymbol{\sigma} = \{\sigma_{u,i} : (u,i) \in \mathcal{D}\}$, and $\sigma_{u,i}$ are independent uniform random variables taking values in $\{-1,+1\}$. The random variables $\sigma_{u,i}$ are called Rademacher variables.

By applying McDiarmid's inequality in Lemma 3, and let $c = \frac{2K_\phi}{|\mathcal{D}|}$, with probability at least $1 - \frac{\eta}{2}$,

$$\left|\mathcal{R}(\mathcal{F}) - \mathbb{E}_{S\sim\mathbb{P}^{|\mathcal{D}|}}\{\mathcal{R}(\mathcal{F})\}\right| \leq 2K_\phi \sqrt{\frac{\log(4/\eta)}{2|\mathcal{D}|}} = K_\phi \sqrt{\frac{2\log(4/\eta)}{|\mathcal{D}|}}.$$

For the rest term $\mathcal{B}(\mathcal{F}) - \underset{S \sim \mathbb{P}^{|\mathcal{D}|}}{\mathbb{E}}[\mathcal{B}(\mathcal{F})]$, by applying McDiarmid's inequality in Lemma 3 and the assumptions that $\hat{p}_{u,i} \geq K_\psi$ and $|e_{u,i} - \tilde{e}_{u,i}| \leq K_\phi$, let $c = \frac{2K_\phi^2 \left(1 + \frac{1}{K_\psi}\right)}{|\mathcal{D}|}$, then with probability at least $1 - \frac{\eta}{2}$,

$$\left| \mathcal{B}(\mathcal{F}) - \underset{S \sim \mathbb{P}^{|\mathcal{D}|}}{\mathbb{E}}[\mathcal{B}(\mathcal{F})] \right| \leq 2K_\phi^2 \left(1 + \frac{1}{K_\psi}\right) \sqrt{\frac{\log(4/\eta)}{2|\mathcal{D}|}} = K_\phi^2 \left(1 + \frac{1}{K_\psi}\right) \sqrt{\frac{2\log(4/\eta)}{|\mathcal{D}|}}.$$

We now bound $\mathcal{B}(\mathcal{F})$ combining the above results. Formally, we have

$$\mathcal{B}(\mathcal{F}) = \underset{S \sim \mathbb{P}^{|\mathcal{D}|}}{\mathbb{E}}[\mathcal{B}(\mathcal{F})] + \left\{ \mathcal{B}(\mathcal{F}) - \underset{S \sim \mathbb{P}^{|\mathcal{D}|}}{\mathbb{E}}[\mathcal{B}(\mathcal{F})] \right\}$$

$$\leq 2K_\phi \left(1 + \frac{1}{K_\psi}\right) \underset{S \sim \mathbb{P}^{|\mathcal{D}|}}{\mathbb{E}}\{\mathcal{R}(\mathcal{F})\} + \left\{ \mathcal{B}(\mathcal{F}) - \underset{S \sim \mathbb{P}^{|\mathcal{D}|}}{\mathbb{E}}[\mathcal{B}(\mathcal{F})] \right\}.$$

With probability at least $1 - \eta$, we have

$$\mathcal{B}(\mathcal{F}) \leq 2K_\phi \left(1 + \frac{1}{K_\psi}\right) \left( \mathcal{R}(\mathcal{F}) + K_\phi \sqrt{\frac{2\log(4/\eta)}{|\mathcal{D}|}} \right) + K_\phi^2 \left(1 + \frac{1}{K_\psi}\right) \sqrt{\frac{2\log(4/\eta)}{|\mathcal{D}|}}$$

$$= K_\phi \left(1 + \frac{1}{K_\psi}\right) \left( 2\mathcal{R}(\mathcal{F}) + K_\phi \sqrt{\frac{18\log(4/\eta)}{|\mathcal{D}|}} \right).$$

We now bound the empirical convariance on the missing outcomes combining the above results. Formally, we have

$$\frac{1}{|\mathcal{D}|} \sum_{(u,i):\, o_{u,i}=0} \frac{\hat{p}_{u,i} - o_{u,i}}{\hat{p}_{u,i}} \cdot (e_{u,i} - \tilde{e}_{u,i}) \leq \left[ \frac{1}{|\mathcal{D}|} \sum_{(u,i) \in \mathcal{D}} (e_{u,i} - \tilde{e}_{u,i})^2 \right]^{\frac{1}{2}}$$

$$\leq \left[ \mathcal{L}_e^{Bal}(\phi, \epsilon) + \frac{K_\phi^2}{|\mathcal{D}|} \sum_{u,i \in \mathcal{D}} \left| 1 - \mathbb{E}\left[ \frac{o_{u,i}}{\hat{p}_{u,i}} \Big| x_{u,i} \right] \right| + K_\phi \left(1 + \frac{1}{K_\psi}\right) \left( 2\mathcal{R}(\mathcal{F}) + K_\phi \sqrt{\frac{18\log(4/\eta)}{|\mathcal{D}|}} \right) \right]^{\frac{1}{2}}$$

$$\leq \mathcal{L}_e^{Bal}(\phi, \epsilon)^{\frac{1}{2}} + K_\phi \cdot \left[ \frac{1}{|\mathcal{D}|} \sum_{u,i \in \mathcal{D}} \left| 1 - \mathbb{E}\left[ \frac{o_{u,i}}{\hat{p}_{u,i}} \Big| x_{u,i} \right] \right| \right]^{\frac{1}{2}} + \left[ K_\phi \left(1 + \frac{1}{K_\psi}\right) \left( 2\mathcal{R}(\mathcal{F}) + K_\phi \sqrt{\frac{18\log(4/\eta)}{|\mathcal{D}|}} \right) \right]^{\frac{1}{2}},$$

which yields the stated results. □

*Remark 1* (Relation to previous imputed errors). The learned coefficient $\epsilon^*$ will converge to zero, when the probabilistic imputation model $\hat{e}_{u,i}$ has already lead to zero empirical covariance, making $\tilde{e}_{u,i}$ degenerates to $\hat{e}_{u,i}$.

*Proof of Remark 1.* Note that the coefficient $\epsilon^*$ is solved by minimizing

$$\frac{1}{|\mathcal{D}|} \sum_{(u,i) \in \mathcal{D}} \frac{o_{u,i}(e_{u,i} - \hat{e}_{u,i} - \epsilon(o_{u,i} - \hat{p}_{u,i}))^2}{\hat{p}_{u,i}}.$$

Taking the first derivative of the above loss with respect to $\epsilon$ and setting it to zero yields

$$\sum_{(u,i) \in \mathcal{D}} \frac{o_{u,i}}{\hat{p}_{u,i}} \cdot \left\{ e_{u,i} - \hat{e}_{u,i} - \epsilon(o_{u,i} - \hat{p}_{u,i}) \right\} \cdot (o_{u,i} - \hat{p}_{u,i}) = 0,$$

which implies that

$$\sum_{(u,i) \in \mathcal{D}} \frac{o_{u,i}}{\hat{p}_{u,i}} \cdot \left\{ e_{u,i} - \tilde{e}_{u,i} \right\} \cdot (o_{u,i} - \hat{p}_{u,i}) = 0,$$

from which implies the uniqueness of $\epsilon$. Formally, if $\hat{e}_{u,i}$ already satisfies zero empirical convariance on the observed outcomes, then $\epsilon = 0$ is a solution of the above equation. Let $\hat{\epsilon}$ is another solution of the above equation. Since the solution of equation is unique, then $\hat{\epsilon}$ will converges to 0. □

*Remark 2* (Bias reduction property). The proposed balancing enhanced imputation loss leads to smaller bias of imputed errors $\tilde{e}_{u,i}$, when $\hat{e}_{u,i}$ are inaccurate. Formally,

$$\min_{\phi,\epsilon} \mathcal{L}_e^{Bal}(\phi,\epsilon) = \frac{1}{|\mathcal{D}|} \sum_{(u,i)\in\mathcal{D}} \frac{o_{u,i}(e_{u,i} - \tilde{e}_{u,i})^2}{\hat{p}_{u,i}} \leq \min_{\phi} \mathcal{L}_e(\phi) = \frac{1}{|\mathcal{D}|} \sum_{(u,i)\in\mathcal{D}} \frac{o_{u,i}(e_{u,i} - \hat{e}_{u,i})^2}{\hat{p}_{u,i}}.$$

*Proof of Remark 2.* The result holds directly by noting that

$$\min_{\phi,\epsilon} \mathcal{L}_e^{Bal}(\phi,\epsilon) \leq \min_{\phi} \mathcal{L}_e^{Bal}(\phi,\epsilon=0) = \min_{\phi} \mathcal{L}_e(\phi) = \frac{1}{|\mathcal{D}|} \sum_{(u,i)\in\mathcal{D}} \frac{o_{u,i}(e_{u,i} - \hat{e}_{u,i})^2}{\hat{p}_{u,i}}.$$

$\square$

*Remark 3* (Variance reduction property). The proposed balancing enhanced imputation loss leads to smaller variance of imputed errors $\tilde{e}_{u,i}$ when the optimal $\epsilon^*$ lies in a certain range. Formally, we have

$$\mathbb{V}(\tilde{e}_{u,i}) = \mathbb{V}(\hat{e}_{u,i} + \epsilon^* \cdot (o_{u,i} - \hat{p}_{u,i})) \leq \mathbb{V}(\hat{e}_{u,i}), \quad \text{if} \quad \epsilon^* \in \left[0, \, 2 \cdot \frac{\text{Cov}(\hat{e}_{u,i}, \hat{p}_{u,i} - o_{u,i})}{\mathbb{V}(\hat{p}_{u,i} - o_{u,i})}\right].$$

*Proof of Remark 3.* The result holds directly by first noting that

$$\mathbb{V}(\tilde{e}_{u,i}) = \mathbb{V}(\hat{e}_{u,i}) - 2\epsilon^* \, \text{Cov}(\hat{e}_{u,i}, \hat{p}_{u,i} - o_{u,i}) + (\epsilon^*)^2 \mathbb{V}(o_{u,i} - \hat{p}_{u,i}),$$

which serves as quadratic function with respect to $\epsilon^*$. By taking the partial derivative with respective to $\epsilon^*$ of the above formula and setting it to zero, the optimal $\epsilon^*$ with the minimal variance is given as

$$\epsilon^* = \frac{\text{Cov}(\hat{e}_{u,i}, \hat{p}_{u,i} - o_{u,i})}{\mathbb{V}(\hat{p}_{u,i} - o_{u,i})}.$$

Therefore, by exploiting the symmetry of the quadratic function, we have

$$\mathbb{V}(\tilde{e}_{u,i}) = \mathbb{V}(\hat{e}_{u,i} + \epsilon^* \cdot (o_{u,i} - \hat{p}_{u,i})) \leq \mathbb{V}(\hat{e}_{u,i}), \quad \text{if} \quad \epsilon^* \in \left[0, \, 2 \cdot \frac{\text{Cov}(\hat{e}_{u,i}, \hat{p}_{u,i} - o_{u,i})}{\mathbb{V}(\hat{p}_{u,i} - o_{u,i})}\right].$$

$\square$

**Theorem 3** (Generalization Bound under Probabilistic Models). *Suppose that $\hat{p}_{u,i} \geq K_\psi$ and $\min\{\hat{e}_{u,i}, |e_{u,i} - \hat{e}_{u,i}|\} \leq K_\phi$, then with probability at least $1 - \eta$, we have*

$$\mathcal{L}_{ideal}(\theta) \leq \mathcal{L}_{DR}(\theta) + \underbrace{\frac{1}{|\mathcal{D}|} \sum_{(u,i)\in\mathcal{D}} \left|1 - \mathbb{E}\left[\frac{o_{u,i}}{\hat{p}_{u,i}}\Big|x_{u,i}\right]\right| \cdot \left|\mathbb{E}[e_{u,i} \mid x_{u,i}] - \mathbb{E}[\hat{e}_{u,i} \mid x_{u,i}]\right|}_{\text{vanilla DR only controls the empirical DR loss, and empirical risks of imputation and propensity models}}$$

$$+ \underbrace{\left|\frac{1}{|\mathcal{D}|} \sum_{(u,i)\in\mathcal{D}} \text{Cov}\left(\frac{o_{u,i} - \hat{p}_{u,i}}{\hat{p}_{u,i}}, e_{u,i} - \hat{e}_{u,i}\right)\right|}_{\text{balancing enhanced DR further controls the independence}} + \underbrace{\left(1 + \frac{1}{K_\psi}\right)\left(2\mathcal{R}(\mathcal{F}) + K_\phi\sqrt{\frac{18\log(4/\eta)}{|\mathcal{D}|}}\right)}_{\text{tail bound controlled by empirical Rademacher complexity and sample size}}$$

*Proof of Theorem 3.* First we decompose the ideal loss as follows.

$$\begin{aligned}
\mathcal{L}_{ideal}(\theta) &= \mathcal{L}_{DR}(\theta) + (\mathcal{L}_{ideal}(\theta) - \mathbb{E}[\mathcal{L}_{DR}(\theta)]) + (\mathbb{E}[\mathcal{L}_{DR}(\theta)] - \mathcal{L}_{DR}(\theta)) \\
&= \mathcal{L}_{DR}(\theta) + \text{Bias}[\mathcal{L}_{DR}(\theta)] + (\mathbb{E}[\mathcal{L}_{DR}(\theta)] - \mathcal{L}_{DR}(\theta)) \\
&\leq \mathcal{L}_{DR}(\theta) + |\text{Bias}[\mathcal{L}_{DR}(\theta)]| \\
&\quad + \sup_{f_\theta \in \mathcal{F}} \left(\mathbb{E}\left[\frac{1}{|\mathcal{D}|} \sum_{(u,i)\in\mathcal{D}} \hat{e}_{u,i} + \frac{o_{u,i}(e_{u,i} - \hat{e}_{u,i})}{\hat{p}_{u,i}}\right] - \frac{1}{|\mathcal{D}|} \sum_{(u,i)\in\mathcal{D}} \hat{e}_{u,i} - \frac{o_{u,i}(e_{u,i} - \hat{e}_{u,i})}{\hat{p}_{u,i}}\right).
\end{aligned}$$

For simplicity, we denote the last term in the above formula as

$$\mathcal{B}(\mathcal{F}) = \sup_{f_\theta \in \mathcal{F}} \left( \mathbb{E}\left[ \frac{1}{|\mathcal{D}|} \sum_{(u,i) \in \mathcal{D}} \hat{e}_{u,i} + \frac{o_{u,i}(e_{u,i} - \hat{e}_{u,i})}{\hat{p}_{u,i}} \right] - \frac{1}{|\mathcal{D}|} \sum_{(u,i) \in \mathcal{D}} \hat{e}_{u,i} - \frac{o_{u,i}(e_{u,i} - \hat{e}_{u,i})}{\hat{p}_{u,i}} \right),$$

we then aim to bound $\mathcal{B}(\mathcal{F})$ in the following.

Note that

$$\mathcal{B}(\mathcal{F}) = \mathbb{E}_{S \sim \mathbb{P}^{|\mathcal{D}|}}[\mathcal{B}(\mathcal{F})] + \left\{ \mathcal{B}(\mathcal{F}) - \mathbb{E}_{S \sim \mathbb{P}^{|\mathcal{D}|}}[\mathcal{B}(\mathcal{F})] \right\},$$

where the first term is $\mathbb{E}_{S \sim \mathbb{P}^{|\mathcal{D}|}}[\mathcal{B}(\mathcal{F})]$, and by Lemma 2 we have

$$\mathbb{E}_{S \sim \mathbb{P}^{|\mathcal{D}|}}[\mathcal{B}(\mathcal{F})] \leq 2 \mathbb{E}_{S \sim \mathbb{P}^{|\mathcal{D}|}} \mathbb{E}_{\boldsymbol{\sigma} \sim \{-1,+1\}^{|\mathcal{D}|}} \sup_{f_\theta \in \mathcal{F}} \left[ \frac{1}{|\mathcal{D}|} \sum_{(u,i) \in \mathcal{D}} \sigma_{u,i} \hat{e}_{u,i} + \frac{\sigma_{u,i} o_{u,i}(e_{u,i} - \hat{e}_{u,i})}{\hat{p}_{u,i}} \right].$$

By the assumptions that $\hat{p}_{u,i} \geq K_\psi$ and $\min\{\hat{e}_{u,i}, |e_{u,i} - \hat{e}_{u,i}|\} \leq K_\phi$, we have

$$\mathbb{E}_{S \sim \mathbb{P}^{|\mathcal{D}|}}[\mathcal{B}(\mathcal{F})] \leq 2 \mathbb{E}_{S \sim \mathbb{P}^{|\mathcal{D}|}} \mathbb{E}_{\boldsymbol{\sigma} \sim \{-1,+1\}^{|\mathcal{D}|}} \sup_{f_\theta \in \mathcal{F}} \left[ \frac{1}{|\mathcal{D}|} \sum_{(u,i) \in \mathcal{D}} \frac{\sigma_{u,i} o_{u,i}(e_{u,i} - \hat{e}_{u,i})}{\hat{p}_{u,i}} \right]$$

$$\leq 2\left(1 + \frac{1}{K_\psi}\right) \mathbb{E}_{S \sim \mathbb{P}^{|\mathcal{D}|}}\{\mathcal{R}(\mathcal{F})\},$$

where the first equation is from Lemma 4, and $\mathcal{R}(\mathcal{F})$ is the empirical Rademacher complexity

$$\mathcal{R}(\mathcal{F}) = \mathbb{E}_{\boldsymbol{\sigma} \sim \{-1,+1\}^{|\mathcal{D}|}} \sup_{f_\theta \in \mathcal{F}} \left[ \frac{1}{|\mathcal{D}|} \sum_{(u,i) \in \mathcal{D}} \sigma_{u,i} e_{u,i} \right],$$

where $\boldsymbol{\sigma} = \{\sigma_{u,i} : (u,i) \in \mathcal{D}\}$, and $\sigma_{u,i}$ are independent uniform random variables taking values in $\{-1, +1\}$. The random variables $\sigma_{u,i}$ are called Rademacher variables.

By applying McDiarmid's inequality in Lemma 3, and let $c = \frac{2K_\phi}{|\mathcal{D}|}$, with probability at least $1 - \frac{\eta}{2}$,

$$\left| \mathcal{R}(\mathcal{F}) - \mathbb{E}_{S \sim \mathbb{P}^{|\mathcal{D}|}}\{\mathcal{R}(\mathcal{F})\} \right| \leq 2K_\phi \sqrt{\frac{\log(4/\eta)}{2|\mathcal{D}|}} = K_\phi \sqrt{\frac{2\log(4/\eta)}{|\mathcal{D}|}}.$$

For the rest term $\mathcal{B}(\mathcal{F}) - \mathbb{E}_{S \sim \mathbb{P}^{|\mathcal{D}|}}[\mathcal{B}(\mathcal{F})]$, by applying McDiarmid's inequality in Lemma 3 and the assumptions that $\hat{p}_{u,i} \geq K_\psi$ and $\min\{\hat{e}_{u,i}, |e_{u,i} - \hat{e}_{u,i}|\} \leq K_\phi$, let $c = \frac{2K_\phi\left(1 + \frac{1}{K_\psi}\right)}{|\mathcal{D}|}$, then with probability at least $1 - \frac{\eta}{2}$,

$$\left| \mathcal{B}(\mathcal{F}) - \mathbb{E}_{S \sim \mathbb{P}^{|\mathcal{D}|}}[\mathcal{B}(\mathcal{F})] \right| \leq 2K_\phi\left(1 + \frac{1}{K_\psi}\right)\sqrt{\frac{\log(4/\eta)}{2|\mathcal{D}|}} = K_\phi\left(1 + \frac{1}{K_\psi}\right)\sqrt{\frac{2\log(4/\eta)}{|\mathcal{D}|}}.$$

We now bound $\mathcal{B}(\mathcal{F})$ combining the above results. Formally, we have

$$\mathcal{B}(\mathcal{F}) = \mathbb{E}_{S \sim \mathbb{P}^{|\mathcal{D}|}}[\mathcal{B}(\mathcal{F})] + \left\{ \mathcal{B}(\mathcal{F}) - \mathbb{E}_{S \sim \mathbb{P}^{|\mathcal{D}|}}[\mathcal{B}(\mathcal{F})] \right\}$$

$$\leq 2\left(1 + \frac{1}{K_\psi}\right) \mathbb{E}_{S \sim \mathbb{P}^{|\mathcal{D}|}}\{\mathcal{R}(\mathcal{F})\} + \left\{ \mathcal{B}(\mathcal{F}) - \mathbb{E}_{S \sim \mathbb{P}^{|\mathcal{D}|}}[\mathcal{B}(\mathcal{F})] \right\}.$$

With probability at least $1 - \eta$, we have

$$\mathcal{B}(\mathcal{F}) \leq 2\left(1 + \frac{1}{K_\psi}\right)\left(\mathcal{R}(\mathcal{F}) + K_\phi\sqrt{\frac{2\log(4/\eta)}{|\mathcal{D}|}}\right) + K_\phi\left(1 + \frac{1}{K_\psi}\right)\sqrt{\frac{2\log(4/\eta)}{|\mathcal{D}|}}$$

$$= \left(1 + \frac{1}{K_\psi}\right)\left(2\mathcal{R}(\mathcal{F}) + K_\phi\sqrt{\frac{18\log(4/\eta)}{|\mathcal{D}|}}\right).$$

We now bound the ideal loss combining the above results. Formally, we have

$$\mathcal{L}_{ideal}(\theta) \leq \mathcal{L}_{DR}(\theta) + |\mathrm{Bias}[\mathcal{L}_{DR}(\theta)]| + \mathcal{B}(\mathcal{F})$$

$$\leq \mathcal{L}_{DR}(\theta) + |\mathrm{Bias}[\mathcal{L}_{DR}(\theta)]| + \left(1 + \frac{1}{K_\psi}\right)\left(2\mathcal{R}(\mathcal{F}) + K_\phi\sqrt{\frac{18\log(4/\eta)}{|\mathcal{D}|}}\right).$$

In Theorem 1, we have already prove that

$$|\mathrm{Bias}[\mathcal{E}_{DR}(\theta)]| = \left| \frac{1}{|\mathcal{D}|}\sum_{(u,i)\in\mathcal{D}}\mathrm{Cov}\left(\frac{\hat{p}_{u,i}-o_{u,i}}{\hat{p}_{u,i}}, e_{u,i}-\hat{e}_{u,i}\right)\right.$$

$$\left. + \frac{1}{|\mathcal{D}|}\sum_{(u,i)\in\mathcal{D}}\left[\left\{1 - \mathbb{E}\left[\frac{o_{u,i}}{\hat{p}_{u,i}}\Big|x_{u,i}\right]\right\}\cdot\left\{\mathbb{E}[e_{u,i}\mid x_{u,i}]-\mathbb{E}[\hat{e}_{u,i}\mid x_{u,i}]\right\}\right] \right|$$

$$\leq \left|\frac{1}{|\mathcal{D}|}\sum_{(u,i)\in\mathcal{D}}\mathrm{Cov}\left(\frac{o_{u,i}-\hat{p}_{u,i}}{\hat{p}_{u,i}}, e_{u,i}-\hat{e}_{u,i}\right)\right|$$

$$+ \frac{1}{|\mathcal{D}|}\sum_{(u,i)\in\mathcal{D}}\left|1 - \mathbb{E}\left[\frac{o_{u,i}}{\hat{p}_{u,i}}\Big|x_{u,i}\right]\right|\cdot\left|\mathbb{E}[e_{u,i}\mid x_{u,i}]-\mathbb{E}[\hat{e}_{u,i}\mid x_{u,i}]\right|,$$

therefore with probability at least $1-\eta$, we have

$$\mathcal{L}_{ideal}(\theta) \leq \mathcal{L}_{DR}(\theta) + \frac{1}{|\mathcal{D}|}\sum_{(u,i)\in\mathcal{D}}\left|1 - \mathbb{E}\left[\frac{o_{u,i}}{\hat{p}_{u,i}}\Big|x_{u,i}\right]\right|\cdot\left|\mathbb{E}[e_{u,i}\mid x_{u,i}]-\mathbb{E}[\hat{e}_{u,i}\mid x_{u,i}]\right|$$

$$+ \left|\frac{1}{|\mathcal{D}|}\sum_{(u,i)\in\mathcal{D}}\mathrm{Cov}\left(\frac{o_{u,i}-\hat{p}_{u,i}}{\hat{p}_{u,i}}, e_{u,i}-\hat{e}_{u,i}\right)\right| + \left(1 + \frac{1}{K_\psi}\right)\left(2\mathcal{R}(\mathcal{F}) + K_\phi\sqrt{\frac{18\log(4/\eta)}{|\mathcal{D}|}}\right),$$

which yields the stated results. $\square$

