# OpenReview forum: "From Deterministic to Probabilistic World: Balancing Enhanced Doubly Robust Learning for Debiased Recommendation"
_ICLR.cc/2024/Conference — Submitted to ICLR 2024_

### Official Review · Reviewer_gTTa · 2023-10-20

**Soundness:** 3 good
**Presentation:** 3 good
**Contribution:** 3 good
**Rating:** 8
**Confidence:** 4

**Summary:**

This paper delves into the realm of doubly robust estimators in recommender systems, specifically focusing on extending deterministic error imputation and propensity models to their probabilistic counterparts. Starting from the derivation of the bias of the doubly robust estimator under probabilistic models, this paper proposes a novel balancing enhanced doubly robust (BEDR) method to control the covariance term in Theorem 1. Furthermore, theoretical analyses on the generalization bound are performed to show the superiority of BEDR. Empirically, experiments on three real-world datasets, including Coat, Yahoo R3 and KuaiRec, are conducted to verify the effectiveness of the proposed method.

**Strengths:**

- This paper is well-written and easy to follow.
- This paper is well-motivated, driven by the transition from deterministic models to probabilistic ones.
- The contribution is solid, supported by detailed theoretical analyses and valuable empirical insights.

**Weaknesses:**

I think this paper can be further improved by addressing the following concerns.
- Empirical insights of advantages of probabilistic models over deterministic ones. Since this paper focuses on extending the deterministic setting to the probabilistic ones, a natural question arises: How do variants of PMF perform in comparison to their MF counterparts, especially when coupled with DR estimators? Thus, I wonder how MF with SOTA DR estimators perform, compared with PMF+BEDR.
- In the experiments, the base model is PMF. However, in practical scenarios, neural network-based MF models like NCF and graph neural network-based MF models such as LightGCN are widely adopted. Hence, it is crucial to incorporate empirical insights into the paper by considering alternative backbones like NCF, LightGCN, etc. This exploration would enrich the experimental findings and align the study more closely with real-world applications.
- Some typos can be fixed by further proof-reading. For example:
    - What is $\mathcal{O}$ in Lemma 1? Is it $\mathbf{O}$? If not, it is not defined.
    - lemma 1 -> Lemma 1, theorem 1 -> Theorem 1, etc.
    - Consider bolding Remark 1-3.

**Questions:**

See the weaknesses.

---

### Official Review · Reviewer_oqbo · 2023-10-25

**Soundness:** 2 fair
**Presentation:** 2 fair
**Contribution:** 2 fair
**Rating:** 3
**Confidence:** 4

**Summary:**

This paper introduces a novel approach to tackling selection bias in recommender systems by extending the Doubly Robust (DR) learning method.\
Unlike traditional DR methods that assume deterministic propensity scores and imputed errors, the authors introduce random propensities and imputed errors. \
The authors develop the Balancing Enhanced Doubly Robust (BEDR) method, which includes a balancing correction term in the imputation model.

**Strengths:**

1. The paper is well-written and thoughtfully organized. Notations are presented clearly, and the preliminary section is well-crafted to facilitate understanding of the manuscript.

2. The theorems and proofs are thorough, and the paper effectively introduces and manages bias and variance. However, there are some problems as noted in Weaknesses.

**Weaknesses:**

1. Motivation
- Why the randomness of learned propensities and imputed errors should be considered?
- I personally cannot understand why there is a correlation between the error of propensity scores and the error of imputed errors.
- How did you implement the random propensity scores and imputed error? A detailed explanation of this process is missing.
- Actually, the proposed empirical covariance term does not have any variance of propensity scores and imputed errors. It even can be computed with deterministic models.

2. Proof of Theorem 1
- I cannot understand some parts of the proof of Theorem 1.
- In the first line, why did you take the expectation of $e_{u,i}$? it is a deterministic value.
- In the second line, why did you use the double expectation formula? Over which variable is each expectation calculated? I think just one expectation is enough.
- This proof is over-complicated. You can just go to the final line from the second line. It is a basic mathematics E[AB] = E[A]E[B]+Cov[A,B].

3. Definition 1
- Cov[A,B] = E[AB] - E[A]E[B].
- However, in Definition 1, the term E[A]E[B] is missing.
- The missing $E[\frac{\hat{p}-o}{\hat{p}}] E[e-\hat{e}]$ becomes zero only when either propensity model or imputed errors are correct.
- However, in practice, these are inaccurate. Even if they are accurate, the covariance term is zero.

4. The balancing correction term
- $\tilde{e} = \hat{e} + \epsilon(o-\hat{p})$.
- In this term, the imputed error is corrected by the error of the propensity model. What is the intuitive concept of this term?
- The derivatives on the $\mathcal{L}_e^{Bal}$ wrt $\epsilon$ should have $\epsilon$ in itself since $\mathcal{L}_e^{Bal}$ has the square of $\epsilon$. I cannot follow this derivative.

**Questions:**

See Weaknesses.

---

### Official Review · Reviewer_XsAT · 2023-10-31

**Soundness:** 3 good
**Presentation:** 3 good
**Contribution:** 3 good
**Rating:** 6
**Confidence:** 2

**Summary:**

This work presents a balancing enhanced doubly robust (BEDR) joint learning approach for unbiased learning under probabilistic error imputations and learned propensities in recommender systems. The authors derive the bias of previous doubly robust (DR) methods and provide alternative unbiasedness conditions for probabilistic error imputations and propensity estimations. The proposed BEDR approach improves the accuracy of imputed errors by adding a balancing correction term to the imputation model, which controls the empirical covariates and reduces bias. The authors also derive the generalization error bound and show that it can be effectively controlled by the proposed learning approach. They conducted experiments on three real-world datasets to demonstrate the effectiveness of the proposed method.

**Strengths:**

The authors provide both theoretical and empirical analysis for the proposed method.

**Weaknesses:**

-

**Questions:**

-

---

### Meta-Review · Area_Chair_jyRs · 2023-12-12

**Metareview:**

The reviewers are not positive about the contribution of the paper, the authors also did not post a rebuttal. So we reject the submission.

**Justification For Why Not Higher Score:**

N/A

**Justification For Why Not Lower Score:**

N/A

---

### Decision · Program_Chairs · 2024-01-16

Reject